# An Empirical Study on the Coupling Coordination and Driving Factors of Rural Revitalization and Rural E-Commerce in the Context of the Digital Economy: The Case of Hunan Province, China

**Canjiang Zhu** and **Weiyi Luo** *

Business College, Hunan University of Humanities, Science and Technology, Loudi 417000, China;
22091002@huhst.edu.cn
* Correspondence: 3275@huhst.edu.cn

**Abstract:** The coupling coordination development of rural revitalization and rural e-commerce is of great significance in promoting the economic growth of rural areas. Based on the observational data of 14 prefectural-level cities in Hunan Province from 2013 to 2021, this study analyzes the spatio-temporal evolution characteristics and driving factors of the coupling coordination development of rural revitalization and rural e-commerce in Hunan Province by adopting the methods of the coupling coordination model and spatial econometric model. The findings indicate that (1) the coupling coordination degree between rural revitalization and rural e-commerce has gradually increased, showing a transition path from dissonance to coordination; (2) there are regional differences in the coupling coordination degree between rural revitalization and rural e-commerce, with the overall difference showing a fluctuating narrowing trend, and the intra-regional difference is the principal cause for the overall difference; (3) the coupling coordination degree shows spatial clustering characteristics, with high–high clustering areas mainly concentrated in the eastern part of Hunan Province and low–low clustering areas concentrated in the western part of Hunan Province; and (4) there is a notable driving effect of infrastructure development, digitalization level, human capital, and regional consumption level on the coupling coordination of rural revitalization and rural e-commerce. Above all, this study provides effective recommendations for promoting the coordinated development of rural revitalization and rural e-commerce.

**Keywords:** rural revitalization; rural e-commerce; coupling coordination degree; driving factors

## 1. Introduction

Jinping Xi, General Secretary of the investigation in Shaanxi in April 2020, has stressed that, as an emerging industry, e-commerce is promising because it is able to market agricultural products, helping rural residents to escape poverty and promoting rural revitalization. Rural e-commerce is an essential manifestation of the fusion of the digital economy and the real economy that not only connects production and markets and links urban and rural regions but also plays a crucial role in facilitating agricultural product sales, expanding farmers' employment, and promoting rural economic growth, thereby injecting a strong impetus into the comprehensive promotion of the rural revitalization strategy [1]. Based on big data from the Ministry of Commerce, China's rural online retail sales reached CNY 2.49 trillion in 2023, increasing by 22.6 times compared with CNY 0.11 trillion in 2013. It can be shown that the rural online retail market has greatly realized the rural consumption potential and activated the rural consumption market, thereby effectively driving economic expansion in rural areas. In other words, to support the realization of rural revitalization to proactively promote rural e-commerce, digital economy development must be considered an opportunity.

In recent years, Hunan Province has voluntarily conformed to the digital economy development trend, having proactively launched a consolidated demonstration project of rural e-commerce and continuously perfected the framework of rural e-commerce public services, and the scale of the rural e-commerce market has been steadily expanding. Data related to the period of the Double 11 shopping festival in 2023 show that rural e-commerce in Hunan Province realized online retail sales of CNY 3.78 billion, up by 14.24% year-on-year. It is evident that, as a new engine of economic growth, rural e-commerce is becoming an essential part of the promotion of rural revitalization. However, from a national perspective, there exists a certain gap between the speed at which Hunan's rural e-commerce market is developing and that of China's eastern coastal cities [2]. Therefore, it is essential to strengthen the research on the maintenance of the existing rural e-commerce market and the expansion of Hunan's rural e-commerce market. In addition, in the digital economy era, rural revitalization provides a broad space for rural e-commerce to grow, while rural e-commerce likewise makes positive contributions to rural revitalization. If the two subsystems can be developed in coupling coordination, it will help to sustainably enhance the endogenous dynamics of rural areas, which is meaningful for promoting the economic development of rural areas. Given this context, importantly, this study clarifies the coupling coordination relationship between rural revitalization and rural e-commerce in Hunan Province and specifies the driving factors for facilitating the coordinated development of rural revitalization and rural e-commerce, meaning it is a greatly valuable reference that could contribute to the rural economic growth of Hunan Province and help to achieve rural revitalization therein.

## 2. Literature Review

### 2.1. Rural Revitalization

The 19th CPC National Congress Report categorically detailed the implementation of the rural revitalization strategy [3]. The rural revitalization strategy is a major decision-making tool of the CPC Central Committee, which has the strategic objective of facilitating socio-economic development and a comprehensive renaissance in rural areas and advancing the rural revitalization strategy in accordance with the general requirements of the indexes of Thriving Industry, Ecological Livability, Rural Civilization, Effective Governance, and Prosperous Life. The implementation of the rural revitalization strategy is an inevitable choice for addressing Chinese society's major contradictions in the new era. For one thing, facilitating the implementation of rural revitalization could narrow the development gap between urban and rural areas and break down the urban–rural dual structure barriers so as to promote the integrated development of urban and rural areas [4]. Rural revitalization has become the key to the development of urban–rural integration by activating the endogenous dynamics of factors such as rural population, land, and industry [5,6]. For another thing, comprehensively promoting rural revitalization is helpful in realizing the common prosperity of all people [7]. The implementation of the rural revitalization strategy helps farmers increase their income and become rich by continuously broadening their income-generating channels, ultimately realizing common prosperity for everyone. In short, rural revitalization has far-reaching significance in resolving the principal contradiction facing Chinese society and realizing the great rejuvenation of the nation of China.

### 2.2. Rural E-Commerce

Rural e-commerce is defined as the use of modern information technologies, such as the Internet and computers, for production and the management of subjects engaged in agriculture-related fields to facilitate the completion of business transactions such as sales, purchases, and the exchange of digital payment for goods and/or services on the Internet [8]. The role of rural e-commerce is crucial in pushing forward rural economic growth, serving as an instrument of the digital economy [9]. Recently, the CPC Central Committee and the State Council have attached great importance to the development of rural e-commerce and vigorously implemented the project of "Digital Commerce Revitalizing

Agriculture". On the one hand, rural e-commerce is a new avenue for selling agricultural products which could effectively help agricultural products to realize the circulation of goods between producers and consumers [10]. Owing to this, as an emerging trade mode, rural e-commerce not only affects economic activities and the social environment but also takes a proactive approach to facilitating information flow, strengthening industrial coordination, and improving market transparency [11]. Hence, it is beneficial to expand rural e-commerce in order to improve the quality and efficiency of the agricultural industry and facilitate the restructuring of the agricultural industry [12]. On the other hand, rural e-commerce is an excellent instrument for alleviating relative poverty in China's rural areas, contributing to increasing rural household incomes and farmers' consumption capacity [13]. At the same time, farmer participation is encouraged in the rural e-commerce development pipeline, as it can enhance farmers' sustainable livelihood capacity and improve their living standards [14]. These findings indicate that rural e-commerce is becoming increasingly critical in facilitating the structural transformation of the agricultural supply side and helping farmers increase their incomes.

### 2.3. Rural Revitalization and Rural E-Commerce

As a core business and typical mode of agricultural and rural digitization, rural e-commerce plays an active part in promoting poverty alleviation and realizing rural revitalization [15]. From the first proposal of the rural revitalization strategy to its full implementation, until now, the results of integrated research on rural revitalization and rural e-commerce have concentrated on the following three aspects. The first is research on the effectiveness of rural e-commerce in enabling rural revitalization. Ali's research [16] showed that e-commerce is a crucial tool for improving the rural population's lifestyle. Especially for the BRICS countries (Brazil, Russia, India, China, and South Africa), rural e-commerce can contribute to structural reforms that help people living in villages and outlying areas to escape poverty and improve their living standards [17]. The second is research on countermeasures for the development of rural e-commerce in the context of the rural revitalization strategy. Huang [18] found that during the course of rural e-commerce development, there existed problems, such as the weak industrial foundation, the deficiencies of e-commerce professionals, the imperfect e-commerce infrastructure, the unclear e-commerce model, the homogenization of products, the lack of services, etc., and put forward corresponding solutions, such as improving the e-commerce infrastructure, strengthening e-commerce manpower training, and realizing the upgrading of products. Wang and Zhang [19] argued that farmers should cultivate positive behavioral attitudes and reduce the negative impact of subjective norms, which can significantly increase farmers' willingness to upgrade e-commerce digitalization so as to contribute to the advancement of rural e-commerce. The third is research exploring the interactive development effect of rural revitalization and rural e-commerce through empirical analyses, but the literature regarding this aspect is less relevant. Guo and Li [20] used a system dynamics model to simulate the effect of the interaction between rural e-commerce and rural revitalization on China's rural economic development from 2010 to 2016 based on statistical data. Feng and Zhang [21] studied 10 rural revitalization demonstration counties in Guizhou Province from 2015 to 2019, and they found that the coupling coordination degree between rural e-commerce and rural revitalization in Guizhou Province was generally on the rise, and the growth rate was higher than the declining rate. In summary, there are many studies based on qualitative methods for analyzing rural revitalization and rural e-commerce, but relatively few studies have explored the interactive development relationship between the two subsystems through using quantitative analysis methods.

The above findings lay a solid foundation for measuring the coupling coordination of rural revitalization and rural e-commerce. However, in the context of the digital economy, the relevant literature suffers from the following shortcomings: (1) The research yardsticks are mostly focused on the macro level, such as national and provincial levels, with fewer research results at the municipal level. But compared with the macro level, the municipal

area, as an important geographical unit at the micro level, is more likely to grow rural e-commerce and achieve rural revitalization. (2) There is a lack of empirical research on the spatio-temporal evolution and regional differences in the coupling coordination relationship between rural revitalization and rural e-commerce in Hunan Province, and the characteristics of the spatio-temporal evolution and driving factors of the coupling coordination relationship between rural revitalization and rural e-commerce have rarely been explored in the context of the digital economy. Given this, this study aims to provide feedback and guidance to boost the coordinated development of rural revitalization and rural e-commerce in Hunan Province through empirical analysis to find out the current situation and driving factors of the development of rural revitalization and rural e-commerce.

## 3. Materials and Methods

### 3.1. Study Area and Data Source

#### 3.1.1. Study Area

Hunan Province is located in the central region of China and is abbreviated as "Xiang". Hunan has a horseshoe-shaped landform surrounded by mountainous terrain on three sides and opening toward the north. Hunan Province has 14 prefectural-level cities (autonomous prefectures), and the capital city is Changsha. As a result of the regional division adjustment, the province has formed four major economic sectors, namely the Chang-Zhu-Tan Area, Southern Hunan Area, Western Hunan Area, and Dongting Lake Area [22]. Among them, the Chang-Zhu-Tan Area refers to the 3 cities of Changsha, Zhuzhou, and Xiangtan; the Southern Hunan Area refers to the 3 cities of Hengyang, Chenzhou, and Yongzhou; the Western Hunan Area refers to the 5 cities of Shaoyang, Zhangjiajie, Huaihua, Loudi, and Xiangxi; and the Dongting Lake Area refers to the 3 cities of Yueyang, Yiyang, and Changde (Figure 1).

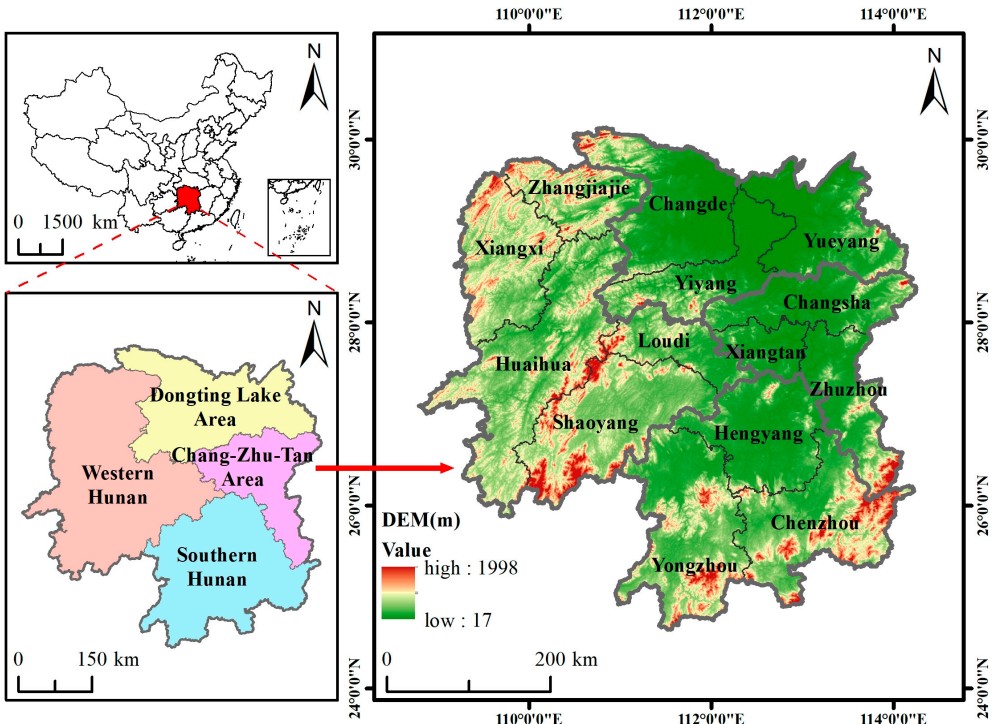

**Figure 1.** Hunan Province map.

#### 3.1.2. Data Source

Given the continuity and availability of sample data, we selected the observations of the 14 prefecture-level cities (autonomous prefectures) in Hunan Province from 2013 to 2021 as the sample for our empirical research. The data used came from the Hunan Provincial Statistical Yearbook, the statistical yearbooks of prefectural cities in Hunan Province, the

Statistical Bulletin of National Economic and Social Development, and the CECN statistical database, among other sources. It should be noted that for specific indicators for which there are missing data for individual years or regions, this study used the average growth rate and the moving average method to estimate the missing data.

*3.2. Methodology*

3.2.1. Entropy Method

The entropy method is a means for identifying index weights which objectively assigns index weights according to the magnitude of index information entropy. When the information entropy of the index is smaller, the more discrete the indicator becomes, with more information contained, the higher the weight assigned [23]. The entropy method avoids the subjectivity bias of subjective assignment methods, which means it is more applicable in comprehensive evaluations. Therefore, this study evaluated the level of rural revitalization and the level of rural e-commerce in Hunan Province using the entropy method.

(1)  Standardization of raw data

Since different indexes have different measurements and units, it is essential to standardize the raw data in order to eliminate the effects of different measurements and units.

Positive index treatment (see Equation (1)):

$$x_{ij}^* = \frac{x_{ij} - \min(x_j)}{\max(x_j) - \min(x_j)}, i \in [1, n], j \in [1, m] \tag{1}$$

Negative index treatment (see Equation (2)):

$$x_{ij}^* = \frac{\max(x_j) - x_{ij}}{\max(x_{ij}) - \min(x_j)}, i \in [1, n], j \in [1, m] \tag{2}$$

where $x_{ij}$ is the value for year i at index j, and $x_{ij}^*$ is the standardized value.

(2)  Calculation of weights and composite scores

Calculate the index share of index j in proportion to the total number of indexes in year i (see Equation (3)):

$$p_{ij} = \frac{x_{ij}^*}{\sum_{i=1}^n x_{ij}^*}, i \in [1, n], j \in [1, m] \tag{3}$$

Calculate the entropy of index j for year i (see Equation (4)):

$$e_j = -\frac{1}{\ln n} \sum_{i=1}^n (p_{ij} \times \ln p_{ij}), i \in [1, n], j \in [1, m] \tag{4}$$

Calculate the information utility value of index j (see Equation (5)):

$$d_j = 1 - e_j, j \in [1, m] \tag{5}$$

Calculate the weight of index j (see Equation (6)):

$$w_j = \frac{d_j}{\sum_{j=1}^m d_j}, j \in [1, m] \tag{6}$$

Calculate the comprehensive score of every city for each year (see Equation (7)):

$$\text{Score}_i = \sum_{j=1}^m (w_j \times x_{ij}^*), i \in [1, n], j \in [1, m] \tag{7}$$

### 3.2.2. Coupling Coordination Degree Model

We used a coupling coordination degree model of rural revitalization and rural e-commerce in Hunan province [24] (see Equation (8)):

$$\begin{aligned}
C &= \frac{2\sqrt{U_1 \times U_2}}{U_1 + U_2} \\
T &= \alpha U_1 + \beta U_2 \\
D &= \sqrt{C \times T}
\end{aligned} \tag{8}$$

where $U_1$ and $U_2$ are the comprehensive scores of rural revitalization and rural e-commerce based on the entropy method of measurement, C is the coupling degree of rural revitalization and rural e-commerce, T is the integrated inter-system coordination index, the values of $\alpha$ and $\beta$ are equally weighted and take the value of 0.5, and D is the coupling coordination degree of the two subsystems, which can reflect the goodness of coordination between the systems. There are 10 levels of coupling coordination results, classified as extreme dysfunction, severe dysfunction, moderate dysfunction, mild dysfunction, verging on dysfunction, barely coordinated, primarily coordinated, intermediately coordinated, good coordination, and high-quality coordination, in which each grade accounts for 10% of the total [25].

### 3.2.3. Thiel Index

The Thiel index indicates regional economic disparities, with higher values indicating greater disparities. Therefore, the Thiel index was used in this study to analyze the inter-regional and intra-regional differences between the two subsystems.

For measuring the total difference in the Thiel index T, Equation (9) is used:

$$T = \frac{1}{n}\sum_i\sum_j \frac{y_{ij}}{\mu} \log \frac{y_{ij}}{\mu} \tag{9}$$

For measuring the total difference in the Thiel index for region i, Equation (10) is used:

$$T_i = \frac{1}{n_i}\sum_j \frac{y_{ij}}{\mu_i} \log \frac{y_{ij}}{\mu_i} \tag{10}$$

The total regional difference in Thiel index is further decomposed into the formulas of intra-regional difference Thiel index $T_{WR}$ and inter-regional difference Thiel index $T_{BR}$, see Equation (11):

$$T = \sum_i \frac{n_i\mu_i}{n\mu}T_i + \frac{1}{n}\sum_i n_i \frac{\mu_i}{\mu} \log \frac{\mu_i}{\mu} = T_{WR} + T_{BR} \tag{11}$$

For the contribution rate calculation, see Equation (12):

$$\begin{aligned}
\text{Contribution Rate of Intra} - \text{Regional Difference} &= \frac{T_{WR}}{T} \times 100\% \\
\text{Contribution Rate of Inter} - \text{Regional Difference} &= \frac{T_{BR}}{T} \times 100\%
\end{aligned} \tag{12}$$

In the above equation, n is the overall number of prefectural-level cities, i represents the region $i = 1, 2, \ldots$, j represents the prefecture-level cities $j = 1, 2, \ldots$, $y_{ij}$ denotes the coupling coordination of prefectural-level city j in region i, $\mu$ represents the mean value of the coupling coordination degree of each prefecture-level city in Hunan Province, $n_i$ represents the count of prefectures in region i, and $\mu_i$ represents the average value of the coupling coordination degree in region i.

### 3.2.4. Moran Index

The famous First Law of Geography states that everything is correlated in a specific way, withe correlations becoming stronger with closer proximity [26]. There are regional differences in the level of coupling coordination between rural revitalization and rural e-commerce in Hunan Province, while the existence of spatial autocorrelation between each prefecture-level city needs to be verified. This study adopted the Moran index to research the spatial relevance of the 14 prefecture-level cities in Hunan Province. The Moran index is divided into the global Moran index and the local Moran index.

The global Moran index can measure the overall geospatial relevance of the coupling coordination of rural revitalization and rural e-commerce in Hunan Province, with a value range of $[-1, 1]$. If the index is positive, it indicates positive spatial autocorrelation, which means that there is spatial agglomeration; if the index is negative, it indicates negative spatial autocorrelation, which means that it is more spatially dispersed; if the index is 0, it denotes no spatial autocorrelation. The formula is as follows:

$$I = \frac{n \sum\limits_{i=1}^{n} \sum\limits_{j=1}^{n} w_{ij}(y_i - \overline{y})(y_j - \overline{y})}{(\sum\limits_{i=1}^{n} \sum\limits_{j=1}^{n} w_{ij}) \sum\limits_{i=1}^{n} (y_i - \overline{y})^2} \tag{13}$$

The local Moran index measures the autocorrelation of the coupling coordination degree within a local region. By measuring the local Moran index, each prefecture-level city can be categorized into four spatial distribution conditions, which are the high–high (HH) type, low–high (LH) type, low–low (LL) type, and high–low (HL) type. Among them, the HH type indicates that the coupling coordination level of the prefecture-level city itself is high and that the level of the surrounding area is also high; the LH type indicates that the coupling coordination level of the prefecture-level city itself is low but the level of the surrounding area is high; the LL type indicates that the coupling coordination level of the prefecture-level city itself is low and the level of the surrounding area is also low; and the HL type indicates that the coupling coordination level of the prefecture-level city itself is high but the level of the surrounding area is low. The formula is as follows:

$$I_i = \frac{n(y_i - \overline{y}) \sum\limits_{j=1, j \neq i}^{n} w_{ij}(y_j - \overline{y})}{\sum\limits_{i=1}^{n} (y_i - \overline{y})^2} \tag{14}$$

In the above equation, n is the total number of regions, $y_i$ is the coupling coordination of region i, $\overline{y}$ is the average of the coupling coordination of the whole province, and $w_{ij}$ is the inverse economic distance spatial weight matrix.

### 3.2.5. Spatial Econometric Model

The level of coupling coordination between rural revitalization and rural e-commerce in Hunan Province is driven by multiple factors. Considering the spatial autocorrelation between the development of rural revitalization and rural e-commerce in Hunan Province, this study introduced spatial econometric models to further explore the driving factors between the two subsystems. For the spatial measurement model equations, see Equation (15):

$$LnD_{it} = \alpha_0 + \rho WLnD_{it} + \theta WLnX_{it} + \beta_0 LnX_{it} + \lambda Wu_{it} + \mu_i + \delta_t + \varepsilon_{it} \tag{15}$$

In the above equation, $D_{it}$ denotes the level of coupling coordination in region i in year t; $X_{it}$ denotes the drivers affecting the coupling coordination of the two subsystems; W is the spatial weight matrix; $\rho$ is the spatial autoregressive coefficient; $\theta$ is the coefficient of the spatial interaction term; $\mu_i$ and $\delta_t$ are the spatial and temporal fixed effects, respectively;

and $\varepsilon_{it}$ is the random error term. If $\lambda = 0$, then it is a spatial Durbin model (SDM); if $\lambda = 0$ and $\theta = 0$, then it is a spatial lag model (SAR); and if $\alpha_0 = \rho = 0$ and $\lambda = 0$, then it is a spatial error model (SEM).

### 3.3. Construction of the Evaluation Index System

3.3.1. Construction of Evaluation Index System for Rural Revitalization in Hunan Province

According to the rich connotations and overall requirements of the rural revitalization strategy, this study selected Thriving Industry, Ecological Livability, Rural Civilization, Effective Governance, and Prosperous Life as first-level indexes [27] and extracted 16 second-level indicators which effectively reflect the current development status of rural revitalization in Hunan Province to construct a rural revitalization evaluation index system, as shown in Table 1. The definitions of the indexes are as follows: Thriving Industry is an essential foundation for rural revitalization. Based on existing research, this study selected four evaluation indexes [21,28], which were used to indicate the degree to which the rural industry is flourishing. Ecological Livability is an inherent requirement of rural revitalization. Based on existing studies, this study selected three evaluation indexes [29,30] which indicate the habitat environments in rural areas. Rural Civilization is the criterion for the advancement and openness of the rural community. Based on existing research, this study selected three evaluation indexes [31] to indicate the farmers' spiritual outlooks. Effective governance is the root of rural revitalization. Based on existing research, the Number of Village Committees was selected as an evaluation indicator [32]. In addition, the Number of Specialized Farmers' Cooperatives and the Number of People with Medical Insurance for Urban and Rural Residents can indirectly reflect the level of rural economic growth and the improvement of farmers' welfare and security in Hunan Province; thus, they were also included in the Effective Governance evaluation index system. Prosperous Life is the starting point and goal of rural revitalization. Based on existing research, this study selected two evaluation indexes [33,34]. In addition, the Number of Automobile Per 100 Households in Rural Areas can characterize farmers' degree of affluence, so it was also included in the Prosperous Life evaluation index system.

**Table 1.** Evaluation indexes for rural revitalization in Hunan Province and their weights.

| Level 1 Indexes | Level 2 Indexes | Unit | Direction | Weight |
|---|---|---|---|---|
| Thriving Industry | Value Added in Primary Industries | CNY 100 Million | Positive | 0.0625 |
| | Total Output Value of Agriculture, Forestry, Animal Husbandry, and Fishery Industries | CNY 10 Thousand | Positive | 0.0651 |
| | Power of Agricultural Machinery | 10,000 KW | Positive | 0.0539 |
| | Total Electricity Consumed by The Rural Society | 10,000 KWH | Positive | 0.0917 |
| Ecological Livability | Consumption of Chemical Fertilizers | 10,000 Tons | Negative | 0.0076 |
| | Afforestation Areas | 10,000 Hectares | Positive | 0.0730 |
| | Access to Sanitary Latrines | % | Positive | 0.0302 |
| Rural Civilization | Proportion of Rural Education and Leisure Consumption Expenditure | % | Positive | 0.0399 |
| | Viewer Rating | % | Positive | 0.0278 |
| | Number of Rural Cultural Stations | Unit | Positive | 0.0435 |
| Effective Governance | Number of Specialized Farmers' Cooperatives | Unit | Positive | 0.1272 |
| | Number of Village Committee | Unit | Positive | 0.1033 |
| | Number of People with Medical Insurance for Urban and Rural Residents | Persons | Positive | 0.1196 |
| Prosperous Life | Per Capita Disposable Income of Rural Households | % | Positive | 0.0741 |
| | Number of Automobile Per 100 Households in Rural Areas | Unit | Positive | 0.0424 |
| | Per Capita Housing Area of Rural Residents | m$^2$ | Positive | 0.0383 |

3.3.2. Construction of Evaluation Index System for Rural E-Commerce in Hunan Province

According to the outer environment of rural e-commerce development in Hunan Province, this study selected Information and Communication Technology [35], Logistics Services [36], Degree of Economic and Social Development [37], and Business Environment [38] as first-level indexes and extracted 16 second-level indicators which reflecting the current situation of rural e-commerce development in Hunan Province to construct a rural e-commerce evaluation index system, as shown in Table 2. The definitions of the indexes are as follows: Information and Communication Technology (ICT) is the means and method for the successful operation of rural e-commerce. Based on existing research, four evaluation indexes were selected in this study to indicate the level of ICT. Logistics Services are a crucial support for rural e-commerce [39]. Based on existing research, this study selected three evaluation indexes to represent logistics and transportation capacity and service level [40,41]. In addition, the Total Length of Fourth Class Highways can reflect the status of basic transportation facilities in rural areas, meaning that this measure has a significant supporting role in the logistics distribution of rural e-commerce; thus, it was also incorporated into the Logistics Services evaluation index system. The extent of economic and social development is closely related to the growth of rural e-commerce [37]. Based on existing studies, four evaluation indexes were selected in this study to indicate the regional economic development status. Business Environment is a crucial foundation for the sustainable growth of rural e-commerce, and this study selected four evaluation indexes to denote the strength of available policy support and market competition [42].

**Table 2.** Evaluation index systems for rural e-commerce in Hunan Province and their weights.

| Level 1 Indexes | Level 2 Indexes | Unit | Direction | Weight |
|---|---|---|---|---|
| Information and communication technology | Revenue from Telecommunication | CNY 100 Million | Positive | 0.0632 |
| | Number of Local Internet Users | 10,000 Households | Positive | 0.0308 |
| | Mobile Telephone Subscribers | 10,000 Households | Positive | 0.0237 |
| | Number of Units in Information Transmission, Computer Service, and Software Industry | Unit | Positive | 0.1168 |
| Logistics Services | Total Length of Fourth Class Highways | km | Positive | 0.0144 |
| | Highway Freight Volume | 10,000 Tons | Positive | 0.0338 |
| | Revenue from Postal Services | CNY 100 Million | Positive | 0.1985 |
| | Number of Units in Transportation, Warehousing and Postal Services | Unit | Positive | 0.0432 |
| Degree of Economic and Social Development | Per Capita GDP | CNY Per Person | Positive | 0.0268 |
| | Per Capita Consumption Expenditure in Rural Areas | CNY | Positive | 0.0146 |
| | Retail Sales of Rural Consumer Goods | CNY 100 Million | Positive | 0.0358 |
| | Workers in The Primary Industries | 10,000 Persons | Negative | 0.0077 |
| Business Environment | E-commerce Transaction Volume | CNY 100 Million | Positive | 0.1202 |
| | Network Retail Sales | CNY 100 Million | Positive | 0.1216 |
| | Proportion of E-Commerce Transaction Volume to GDP | % | Positive | 0.0437 |
| | Demonstrable integration of E-Commerce into Rural Counties | Unit | Positive | 0.1052 |

## 4. Empirical Analysis

### 4.1. Rural Revitalization Development Index

As shown in Table 3, from a time perspective, the average value of the Rural Revitalization Development Index in Hunan Province keeps on increasing, as it increased from 0.270 in 2013 to 0.478 in 2021, which demonstrates that rural revitalization efforts in Hunan Province have been effective and that rural revitalization levels have obviously improved. In subregional terms, the values of the Rural Revitalization Development Index varied among economic regions during the sample examination period. Among them, the

Southern Hunan Area has the highest value (0.464) and has been in the leading position in Hunan Province for a long time, the Dongting Lake Area is in second (0.425), the Chang-Zhu-Tan Area is in third (0.391), and the Western Hunan Area has the lowest value (0.315), which indicates that the level of rural revitalization in Hunan Province shows a spatial distribution structure of "high in the east and low in the west".

**Table 3.** Rural Revitalization Development Index values for Hunan Province (2013–2021).

| Region | City and State | 2013 | 2014 | 2015 | 2016 | 2017 | 2018 | 2019 | 2020 | 2021 | Time Average |
|---|---|---|---|---|---|---|---|---|---|---|---|
| Chang-Zhu-Tan Area | Changsha | 0.453 | 0.491 | 0.511 | 0.519 | 0.600 | 0.587 | 0.668 | 0.637 | 0.572 | 0.560 |
| | Zhuzhou | 0.227 | 0.264 | 0.278 | 0.257 | 0.332 | 0.422 | 0.408 | 0.452 | 0.410 | 0.339 |
| | Xiangtan | 0.165 | 0.208 | 0.239 | 0.257 | 0.232 | 0.333 | 0.344 | 0.396 | 0.286 | 0.273 |
| | Regional Average | 0.281 | 0.321 | 0.343 | 0.344 | 0.388 | 0.447 | 0.474 | 0.495 | 0.423 | 0.391 |
| Southern Hunan Area | Hengyang | 0.385 | 0.450 | 0.458 | 0.482 | 0.598 | 0.541 | 0.617 | 0.674 | 0.589 | 0.533 |
| | Chenzhou | 0.291 | 0.316 | 0.282 | 0.317 | 0.385 | 0.406 | 0.483 | 0.465 | 0.471 | 0.380 |
| | Yongzhou | 0.316 | 0.380 | 0.394 | 0.409 | 0.529 | 0.574 | 0.543 | 0.602 | 0.566 | 0.479 |
| | Regional Average | 0.331 | 0.382 | 0.378 | 0.403 | 0.504 | 0.507 | 0.548 | 0.581 | 0.542 | 0.464 |
| Western Hunan Area | Shaoyang | 0.341 | 0.408 | 0.404 | 0.379 | 0.475 | 0.564 | 0.627 | 0.585 | 0.625 | 0.490 |
| | Zhangjiajie | 0.089 | 0.140 | 0.124 | 0.177 | 0.220 | 0.188 | 0.247 | 0.195 | 0.212 | 0.177 |
| | Huaihua | 0.286 | 0.279 | 0.279 | 0.301 | 0.390 | 0.388 | 0.437 | 0.385 | 0.408 | 0.350 |
| | Loudi | 0.232 | 0.243 | 0.260 | 0.318 | 0.389 | 0.329 | 0.408 | 0.363 | 0.408 | 0.328 |
| | Xiangxi | 0.146 | 0.145 | 0.180 | 0.143 | 0.250 | 0.267 | 0.305 | 0.293 | 0.350 | 0.231 |
| | Regional Average | 0.219 | 0.243 | 0.250 | 0.264 | 0.344 | 0.347 | 0.405 | 0.364 | 0.401 | 0.315 |
| Dongting Lake Area | Yueyang | 0.312 | 0.329 | 0.306 | 0.355 | 0.413 | 0.452 | 0.497 | 0.575 | 0.552 | 0.421 |
| | Changde | 0.349 | 0.304 | 0.397 | 0.431 | 0.481 | 0.483 | 0.497 | 0.563 | 0.570 | 0.453 |
| | Yiyang | 0.276 | 0.273 | 0.319 | 0.306 | 0.402 | 0.436 | 0.474 | 0.567 | 0.557 | 0.401 |
| | Regional Average | 0.312 | 0.302 | 0.340 | 0.364 | 0.432 | 0.457 | 0.489 | 0.568 | 0.560 | 0.425 |
| Whole Province | Regional Average | 0.270 | 0.289 | 0.302 | 0.321 | 0.401 | 0.416 | 0.459 | 0.470 | 0.478 | 0.379 |

At the municipal level, each city has a different value based on the Rural Revitalization Development Index, with these values being calculated using data from the period from 2013 to 2021. Among them, Changsha has the highest average value (0.560), and Zhangjiajie has the lowest average value (0.177). In addition, the Rural Revitalization Development Index has increased in all cities, but there are differences in the speed and magnitude of growth, with Zhangjiajie recording the greatest average annual growth rate of 11.1 percent. Zhangjiajie shows a strong catch-up effect, which is thanks to the development of the tourism industry in Zhangjiajie, according to local conditions, and efforts to promote the Tianmen Mountain–Wulingyuan scenic spot as the core tourist attraction, which have brought benefits that have radiated to the countryside and helped to incorporate farmers into the tourism industry chain, thus causing farmers to become rich, which, in turn, promotes rural industrial revitalization.

### 4.2. Rural E-Commerce Development Index

As shown in Table 4, from a time perspective, the average rural e-commerce development index value in Hunan Province mainly shows an increasing tendency, from 0.039 in 2013 to 0.109 in 2021, indicating that rural e-commerce in Hunan Province is in good shape. The average value of the rural e-commerce development index of Hunan Province for 2020 is 0.249, which is abnormally high. This is due to the fact that, for one thing, the offline retail industry was hurt by the epidemic, which has led to many merchants turning towards online operations. At the same time, the rise of the "home economy" has prompted consumers to make online purchases, normalizing these purchases. Furthermore, the postal service launched the "Express to the Village" project in 2020, which ensures that agricultural products are sold to cities and that industrial products are sold to rural areas.

**Table 4.** Rural e-commerce development index values for Hunan Province (2013–2021).

| Region | City and State | 2013 | 2014 | 2015 | 2016 | 2017 | 2018 | 2019 | 2020 | 2021 | Time Average |
|---|---|---|---|---|---|---|---|---|---|---|---|
| Chang-Zhu-Tan Area | Changsha | 0.172 | 0.176 | 0.262 | 0.317 | 0.408 | 0.504 | 0.615 | 0.799 | 0.578 | 0.426 |
| | Zhuzhou | 0.060 | 0.058 | 0.100 | 0.093 | 0.130 | 0.127 | 0.101 | 0.224 | 0.134 | 0.114 |
| | Xiangtan | 0.040 | 0.039 | 0.071 | 0.058 | 0.067 | 0.080 | 0.126 | 0.221 | 0.142 | 0.094 |
| | Regional Average | 0.090 | 0.091 | 0.144 | 0.156 | 0.202 | 0.237 | 0.281 | 0.415 | 0.285 | 0.211 |
| Southern Hunan Area | Hengyang | 0.046 | 0.045 | 0.052 | 0.060 | 0.074 | 0.089 | 0.133 | 0.299 | 0.143 | 0.105 |
| | Chenzhou | 0.052 | 0.053 | 0.065 | 0.112 | 0.109 | 0.141 | 0.116 | 0.277 | 0.118 | 0.116 |
| | Yongzhou | 0.042 | 0.035 | 0.062 | 0.071 | 0.084 | 0.099 | 0.092 | 0.300 | 0.113 | 0.100 |
| | Regional Average | 0.047 | 0.044 | 0.060 | 0.081 | 0.089 | 0.110 | 0.114 | 0.292 | 0.125 | 0.107 |
| Western Hunan Area | Shaoyang | 0.035 | 0.038 | 0.068 | 0.073 | 0.125 | 0.141 | 0.121 | 0.314 | 0.100 | 0.113 |
| | Zhangjiajie | 0.018 | 0.015 | 0.019 | 0.022 | 0.070 | 0.033 | 0.049 | 0.103 | 0.047 | 0.042 |
| | Huaihua | 0.029 | 0.031 | 0.037 | 0.087 | 0.118 | 0.169 | 0.076 | 0.213 | 0.075 | 0.093 |
| | Loudi | 0.032 | 0.032 | 0.058 | 0.041 | 0.098 | 0.068 | 0.081 | 0.233 | 0.080 | 0.080 |
| | Xiangxi | 0.014 | 0.016 | 0.019 | 0.024 | 0.072 | 0.144 | 0.047 | 0.180 | 0.066 | 0.065 |
| | Regional Average | 0.026 | 0.026 | 0.040 | 0.049 | 0.097 | 0.111 | 0.075 | 0.208 | 0.073 | 0.078 |
| Dongting Lake Area | Yueyang | 0.061 | 0.067 | 0.091 | 0.078 | 0.111 | 0.110 | 0.119 | 0.285 | 0.187 | 0.123 |
| | Changde | 0.053 | 0.055 | 0.065 | 0.094 | 0.105 | 0.102 | 0.117 | 0.273 | 0.124 | 0.110 |
| | Yiyang | 0.039 | 0.039 | 0.052 | 0.081 | 0.079 | 0.110 | 0.100 | 0.255 | 0.126 | 0.098 |
| | Regional Average | 0.051 | 0.054 | 0.069 | 0.085 | 0.098 | 0.107 | 0.112 | 0.271 | 0.145 | 0.110 |
| Whole Province | Regional Average | 0.039 | 0.040 | 0.056 | 0.070 | 0.098 | 0.113 | 0.097 | 0.249 | 0.109 | 0.097 |

In terms of regions, the average value of the rural e-commerce development index in different regions showed a large difference during the sample examination period. The Chang-Zhu-Tan Area has the highest value (0.211), the Dongting Lake Area is in second (0.110), the Southern Hunan Area is in third (0.107), and the Western Hunan Area has the lowest value (0.078), which indicates that the level of rural e-commerce development in Hunan Province displays a spatial distribution structure of "high in the east and low in the west".

At the municipal level, the mean value of the rural e-commerce development index varied across municipalities from 2013 to 2021. Changsha has the highest mean value (0.426), and Zhangjiajie has the lowest mean value (0.042). In addition, the rural e-commerce development index value for Xiangxi has the largest average annual growth rate of 56.6% due to the fact that Xiangxi has high-profile regional public brands such as "Xiangxi Kiwifruit" and "Shibadong Village", and the Xiangxi government has vigorously improved the region's business environment and provided increased financial subsidies to the region, which has further accelerated rural e-commerce development in Xiangxi.

### 4.3. Analysis of the Coupling Coordination between Rural Revitalization and Rural E-Commerce

We analyzed the development trends regarding the coupling coordination degree between rural revitalization and rural e-commerce in Hunan Province's 14 cities from 2013 to 2021 based on the coupling coordination degree model (see Figure 2). As can be seen from Table 5, the mean coupling coordination degree value showed a sustained upward trend during the study period, rising from 0.329 in 2013 to 0.490 in 2021. This indicates that there exists a dynamic coupling coordination development relationship and an increasing degree of coordination between the two subsystems, showing an evolutionary path from dysfunctionality to coordination. The mean value of coupling coordination for 2020 is abnormally high, which is due to the higher value of rural e-commerce development index in that year, causing the higher value of coupling coordination.

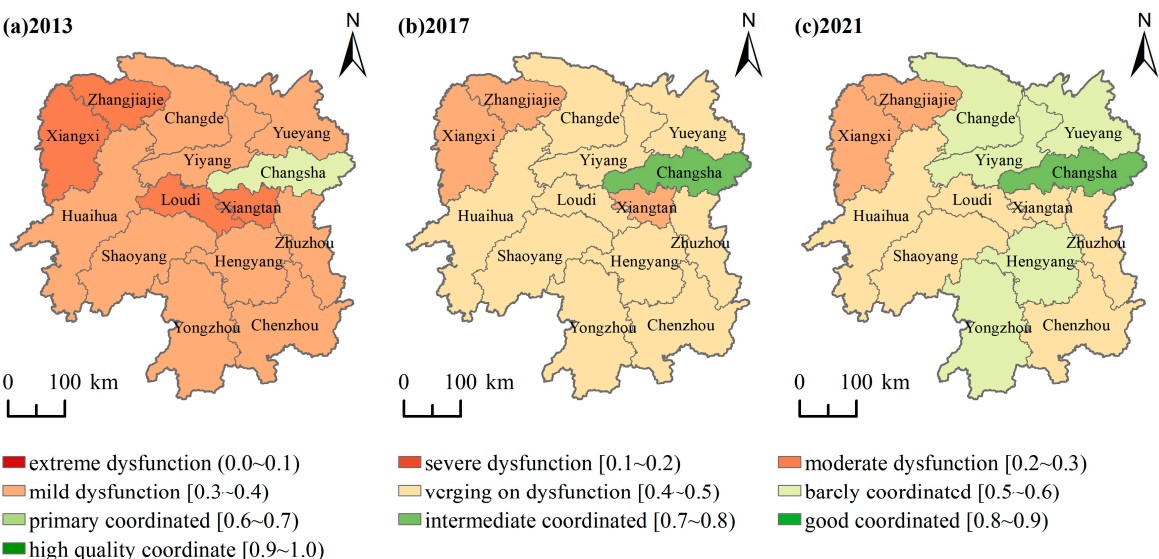

**Figure 2.** The coupling coordination degree between rural revitalization and rural e-commerce in Hunan Province: (**a**) 2013, (**b**) 2017, and (**c**) 2021.

**Table 5.** Coupling coordination degree of rural revitalization and rural e-commerce in Hunan Province (2013–2021).

| Region | City and State | 2013 | 2014 | 2015 | 2016 | 2017 | 2018 | 2019 | 2020 | 2021 | Time Average |
|---|---|---|---|---|---|---|---|---|---|---|---|
| Chang-Zhu-Tan Area | Changsha | 0.528 | 0.542 | 0.605 | 0.637 | 0.703 | 0.737 | 0.801 | 0.845 | 0.758 | 0.684 |
| | Zhuzhou | 0.341 | 0.352 | 0.408 | 0.393 | 0.456 | 0.481 | 0.451 | 0.564 | 0.484 | 0.437 |
| | Xiangtan | 0.285 | 0.301 | 0.361 | 0.350 | 0.354 | 0.404 | 0.457 | 0.544 | 0.449 | 0.389 |
| | Regional Average | 0.385 | 0.398 | 0.458 | 0.460 | 0.504 | 0.541 | 0.569 | 0.651 | 0.564 | 0.503 |
| Southern Hunan Area | Hengyang | 0.365 | 0.377 | 0.393 | 0.412 | 0.458 | 0.468 | 0.535 | 0.670 | 0.539 | 0.469 |
| | Chenzhou | 0.351 | 0.359 | 0.368 | 0.434 | 0.452 | 0.489 | 0.486 | 0.599 | 0.486 | 0.447 |
| | Yongzhou | 0.339 | 0.339 | 0.396 | 0.412 | 0.459 | 0.489 | 0.473 | 0.652 | 0.503 | 0.451 |
| | Regional Average | 0.352 | 0.359 | 0.386 | 0.420 | 0.456 | 0.482 | 0.498 | 0.640 | 0.509 | 0.456 |
| Western Hunan Area | Shaoyang | 0.331 | 0.352 | 0.407 | 0.407 | 0.494 | 0.531 | 0.525 | 0.654 | 0.500 | 0.467 |
| | Zhangjiajie | 0.201 | 0.213 | 0.221 | 0.251 | 0.352 | 0.281 | 0.332 | 0.376 | 0.316 | 0.282 |
| | Huaihua | 0.301 | 0.305 | 0.320 | 0.402 | 0.463 | 0.506 | 0.426 | 0.535 | 0.418 | 0.408 |
| | Loudi | 0.293 | 0.296 | 0.351 | 0.339 | 0.442 | 0.387 | 0.427 | 0.539 | 0.425 | 0.389 |
| | Xiangxi | 0.211 | 0.221 | 0.243 | 0.243 | 0.366 | 0.443 | 0.347 | 0.479 | 0.389 | 0.327 |
| | Regional Average | 0.267 | 0.277 | 0.308 | 0.328 | 0.423 | 0.429 | 0.411 | 0.517 | 0.410 | 0.375 |
| Dongting Lake Area | Yueyang | 0.371 | 0.385 | 0.408 | 0.408 | 0.463 | 0.473 | 0.493 | 0.636 | 0.567 | 0.467 |
| | Changde | 0.369 | 0.360 | 0.401 | 0.449 | 0.474 | 0.471 | 0.491 | 0.626 | 0.515 | 0.462 |
| | Yiyang | 0.322 | 0.321 | 0.359 | 0.396 | 0.422 | 0.468 | 0.467 | 0.617 | 0.514 | 0.432 |
| | Regional Average | 0.354 | 0.355 | 0.389 | 0.418 | 0.453 | 0.470 | 0.484 | 0.626 | 0.532 | 0.454 |
| Whole Province | Regional Average | 0.329 | 0.337 | 0.374 | 0.395 | 0.454 | 0.473 | 0.479 | 0.595 | 0.490 | 0.437 |

There is a difference in the average values of the coupling coordination of rural revitalization and rural e-commerce among the four major economic sectors. Among them, the Chang-Zhu-Tan Area has the highest mean value (0.503), the Southern Hunan Area has the second-highest mean value (0.456), the Dongting Lake Area has the third-highest mean value (0.454), and the Western Hunan Area has the lowest mean value (0.375). In addition, during the research period, all four major economic regions in Hunan Province showed a noticeable upward trend in the degree of coupling coordination, with the type of coupling coordination degree of the Chang-Zhu-Tan Area, Southern Hunan Area, and Dongting Lake Area changing from mild dysfunction to barely coordinated and the type of coupling coordination degree of the Western Hunan Area changing from moderate dysfunction to verging on dysfunction.

At the municipal level, the coupling coordination of rural revitalization and rural e-commerce in the prefecture-level cities in Hunan Province shows an overall upward trend, but there is a difference among the prefecture-level cities. Compared with the other prefecture-level cities in Hunan Province during the same period, Changsha kept a high level of coupling coordination, rising from barely coordinated in 2013 to intermediate coordinated in 2021. In contrast, Zhangjiajie had the lowest level of coupling coordination, rising from moderate dysfunction in 2013 to mild dysfunction in 2021, indicating long-term dysfunction; Xiangxi also showed chronically low-level coupling, rising from moderate dysfunction in 2013 to mild dysfunction in 2021, and its coupling coordination is higher than that of Zhangjiajie. In addition, Hengyang, Yongzhou, Shaoyang, Yueyang, Changde, and Yiyang improved from mild dysfunction in 2013 to barely coordinated in 2021, while Zhuzhou, Chenzhou, Shaoyang, and Huaihua improved from mild dysfunction in 2013 to verging on dysfunction in 2021, and Xiangtan and Loudi improved from moderate dysfunction in 2013 to verging on dysfunction in 2021. It can be seen that Changsha occupies an absolute leading position, while Zhangjiajie and Xiangxi are in the low-value range, with the effect of spatial locking.

*4.4. Regional Difference in the Coupling Coordination Degree of Rural Revitalization and Rural E-Commerce*

After analyzing the coupling coordination degree of rural revitalization and rural e-commerce in Hunan Province, this study found that the development of the eastern areas of Hunan Province is better than that of the western areas, and the development status of the four major regions within Hunan Province is different. Therefore, the Thiel index can be used to analyze the relative differences and evolutionary trends of the coupling coordination development of the four major regions in Hunan Province. Table 6 demonstrates the Thiel index and contribution rate of the coupling coordination degree of rural revitalization and rural e-commerce in Hunan Province. Regarding the total difference, the Thiel index value of the coupling coordination between rural revitalization and rural e-commerce in Hunan Province decreased from 0.0258 in 2013 to 0.0191 in 2021, with slight fluctuations during this period, showing a downward trend, which indicates a fluctuating and narrowing trend regarding the total difference in the level of coupling coordination. It can be seen that, for one thing, the national policy inclination and rural revitalization strategy have remarkable effectiveness. For another thing, the construction of logistic nets, the promotion of digital payment, the popularization of information technology, and other measures designed to boost rural e-commerce to help the countryside's economic growth have driven the coordinated development of rural revitalization and rural e-commerce.

Regarding the decomposition results, the contribution rate of intra-regional difference from 2013 to 2021 is more than 50%, which means the intra-regional difference is greater than the inter-region difference, showing that the total difference in the coupling coordination degree of rural revitalization and rural e-commerce in Hunan Province mainly comes from the great intra-regional difference. The reason for this problem is mainly that Hunan Province tends to give resources, policies, funds, and other development factors to certain cities or counties when boosting the development of rural revitalization and rural e-commerce, as evidenced by the implementation of pilot areas for rural revitalization and the selection of comprehensive demonstration counties for e-commerce in the countryside. Particularly, it was noted that the inter-regional contribution rate in 2017–2018 was significantly reduced due to the massive flooding throughout Hunan Province in the summer of 2017, which caused heavy damage to crops and houses and, thus, led to a reduction in the inter-regional differences in the coordinated development of rural revitalization and rural e-commerce.

Regarding the decomposition results pertaining to the intra-regional differences, the average values of the Thiel index of the coupling coordination degree of rural revitalization and rural e-commerce in the Chang-Zhu-Tan Area, Southern Hunan Area, Western Hunan Area, and Dongting Lake Area in 2013–2021 are 0.0327, 0.0007, 0.0176, and 0.0012,

respectively, while the average values of the contribution rate are 39.18%, 0.72%, 25.03%, and 1.17%, respectively, which indicates that the difference and contribution rate of the Chang-Zhu-Tan Area in Hunan Province is the largest, followed by the Western Hunan Area, and the values of the Southern Hunan Area and Dongting Lake Area are very small. The reason for this is that the development of the prefecture-level cities in Hunan Province is not balanced, and the large differences within the Chang-Zhu-Tan Area are related to the extreme prominence of Changsha, while the big differences within the Western Hunan Area have mostly been caused by the subsidies given to Zhangjiajie and Xiangxi.

**Table 6.** Thiel index and contribution rate of coupled coordination degree of rural revitalization and rural e-commerce in Hunan Province (2013–2021).

| Year | Total Difference | Inter-Region Difference and Contribution Rate | Intra-Region Difference and Contribution Rate | | | | |
|---|---|---|---|---|---|---|---|
| | | | Total | Chang-Zhu-Tan Area | Southern Hunan Area | Western Hunan Area | Dongting Lake Area |
| 2013 | 0.0258 | 0.0107 (41.48) | 0.0151 (59.52) | 0.0354 (34.40) | 0.0004 (0.38) | 0.0194 (21.84) | 0.0021 (1.89) |
| 2014 | 0.0246 | 0.0100 (40.77) | 0.0146 (59.19) | 0.0328 (33.73) | 0.0010 (0.89) | 0.0185 (22.05) | 0.0028 (2.53) |
| 2015 | 0.0257 | 0.0112 (43.53) | 0.0145 (56.47) | 0.0256 (26.13) | 0.0005 (0.45) | 0.0250 (28.54) | 0.0016 (1.35) |
| 2016 | 0.0254 | 0.0089 (35.13) | 0.0165 (64.87) | 0.0361 (35.44) | 0.0003 (0.28) | 0.0239 (27.86) | 0.0015 (1.29) |
| 2017 | 0.0151 | 0.0021 (14.01) | 0.0130 (85.99) | 0.0413 (65.25) | 0.00002 (0.03) | 0.0086 (18.92) | 0.0012 (1.76) |
| 2018 | 0.0194 | 0.0037 (19.11) | 0.0157 (80.89) | 0.0336 (42.32) | 0.0002 (0.24) | 0.0230 (38.34) | 0.00001 (0.01) |
| 2019 | 0.0221 | 0.0075 (33.97) | 0.0146 (66.08) | 0.0392 (45.02) | 0.0015 (1.46) | 0.0139 (19.30) | 0.0003 (0.29) |
| 2020 | 0.0152 | 0.0051 (33.29) | 0.0101 (66.71) | 0.0214 (33.04) | 0.0011 (1.70) | 0.0156 (31.85) | 0.0001 (0.12) |
| 2021 | 0.0191 | 0.0083 (43.73) | 0.0107 (56.27) | 0.0289 (37.30) | 0.0009 (1.10) | 0.0106 (16.60) | 0.0011 (1.28) |

Note: Values in parentheses are contribution rate, units in %.

*4.5. Spatial Distribution of Coupling Coordination Degree of Rural Revitalization and Rural E-Commerce*

Table 7 shows the global Moran index values pertaining to the coupling coordination of rural revitalization and rural e-commerce in Hunan Province from 2013 to 2021, which were calculated using the inverse economic distance weight matrix, reflecting the spatial correlation of the coupling coordination level of the 14 prefectural-level cities in Hunan Province. As shown in Table 7, the global Moran index values are all greater than 0, and the *p*-value is smaller than 0.1 in the majority of the years. This indicates that the coupling coordination degree of the two subsystems between prefectural-level cities in Hunan Province shows a positive spatial autocorrelation with a spatial clustering effect and that it is also affected by the neighboring prefectural-level cities.

**Table 7.** Global Moran index of coupling coordination degree of rural revitalization and rural e-commerce in Hunan Province (2013–2021).

| Year | Global Moran Index | *p*-Value | Year | Global Moran Index | *p*-Value |
|---|---|---|---|---|---|
| 2013 | 0.187 | 0.034 | 2018 | 0.012 | 0.254 |
| 2014 | 0.187 | 0.032 | 2019 | 0.140 | 0.040 |
| 2015 | 0.191 | 0.029 | 2020 | 0.199 | 0.032 |
| 2016 | 0.149 | 0.055 | 2021 | 0.137 | 0.064 |
| 2017 | 0.056 | 0.143 | | | |

Table 8 shows the spatial distribution of the local Moran index from 2013 to 2021. Similarly, the coupling coordination degree of rural revitalization and rural e-commerce in Hunan Province shows a significant spatial correlation. The prefectural-level cities in which the HH-type region occurs are all in the eastern part of Hunan Province, mainly including seven cities: Changsha, Yueyang, Changde, Hengyang, Chenzhou, Yongzhou, and Shaoyang. Among them, Changde and Yueyang in the Dongting Lake Area, Hengyang in the Southern Hunan Area, and Shaoyang in the Western Hunan Area have appeared for nine consecutive years; Chenzhou and Yongzhou in the Southern Hunan Area have appeared for eight years; and Chang-Zhu-Tan Area has appeared for seven years. This type of region is usually characterized by better geography, a favorable location, good resource endowment, and a long history of development, all of which are advantages conducive to the expansion of the rural economy. Zhangjiajie and Xiangxi are perennially located in the LL-type region, which has a wide area of mountainous areas and poor infrastructure development, especially in terms of the low level of transportation access, which has become an important factor hindering rural revitalization and rural e-commerce. Zhuzhou was found to be among the cities characterized as HL-type regions for the period 2013–2018 because its neighboring city is Xiangtan, which has a lower level of coupling coordination than that of Zhuzhou, and for the period 2020–2021, Yiyang was identified because its neighboring city is Loudi, which has a lower level of coupling coordination than that of Yiyang. Xiangtan and Loudi fall into the LH-type region category in most years as a result of the higher level of coupling coordination of rural revitalization and rural e-commerce in their neighboring regions.

**Table 8.** Spatial distribution of local Moran index (2013–2021).

| Year | HH | LH | LL | HL |
|---|---|---|---|---|
| 2013 | Changsha, Yueyang, Changde, Hengyang, Chenzhou, Yongzhou, Shaoyang | Xiangtan, Loudi | Huaihua, Yiyang, Zhangjiajie, Xiangxi | Zhuzhou |
| 2014 | Changsha, Yueyang, Changde, Hengyang, Chenzhou, Yongzhou, Shaoyang | Xiangtan | Loudi, Huaihua, Yiyang, Zhangjiajie, Xiangxi | Zhuzhou |
| 2015 | Changsha, Yueyang, Changde, Hengyang, Yongzhou, Shaoyang | Yiyang, Xiangtan, Chenzhou, | Loudi, Huaihua, Zhangjiajie, Xiangxi | Zhuzhou |
| 2016 | Changsha, Yueyang, Changde, Hengyang, Chenzhou, Yongzhou, Shaoyang | Loudi | Xiangtan, Zhangjiajie, Xiangxi | Zhuzhou, Huaihua, Yiyang, |
| 2017 | Yueyang, Changde, Hengyang, Chenzhou, Yongzhou, Shaoyang | Xiangtan, Yiyang | Loudi, Zhangjiajie, Xiangxi | Changsha, Huaihua, Zhuzhou |
| 2018 | Yueyang, Changde, Hengyang, Chenzhou, Yongzhou, Shaoyang, Huaihua | Loudi, Xiangtan | Yiyang, Zhangjiajie, Xiangxi | Changsha, Zhuzhou |
| 2019 | Changsha, Yueyang, Changde, Hengyang, Chenzhou, Shaoyang | Yongzhou | Loudi, Xiangtan, Zhuzhou, Huaihua, Yiyang, Zhangjiajie, Xiangxi | Lack |
| 2021 | Changsha, Yueyang, Changde, Hengyang, Chenzhou, Yongzhou, Shaoyang | Loudi | Xiangtan, Zhuzhou, Huaihua, Zhangjiajie, Xiangxi | Yiyang |
| 2021 | Changsha, Yueyang, Changde, Hengyang, Chenzhou, Yongzhou, Shaoyang | Loudi, Xiangtan | Zhuzhou, Huaihua, Zhangjiajie, Xiangxi | Yiyang |

In summary, the analysis shows that the distribution trend of the level of coupling coordination of rural revitalization and rural e-commerce in Hunan Province is gradually strengthening from west to east. However, different kinds of elements affect the coordinated development of the eastern and western regions of Hunan Province. Hence, it is crucial to investigate the driving factors affecting the coupling coordination of rural revitalization

and rural e-commerce in Hunan Province to help the coordinated development of the two subsystems.

*4.6. Analysis of Driving Factors for Coupled and Coordinated Development of Rural Revitalization and Rural E-Commerce in Hunan Province*

4.6.1. Variable Selection and Data Description

We captured the driving factors of coupling coordination development between rural revitalization and rural e-commerce in Hunan Province. For one thing, this study combed the literature on the influential factors of rural revitalization and rural e-commerce in recent years and found that resource endowment [43], economic location [43], policy factors [44], and infrastructure [44] are the fundamental drivers affecting rural revitalization and that Internet penetration [42], rural elites [45], government support [45], logistics [46], and agricultural industry structure [46] play an influential role in contributing to the construction of the rural e-commerce industry chain. For another thing, based on the weights of each evaluation index measured in the previous section, this study found that in terms of realizing rural revitalization, the Number of Specialized Farmers' Cooperatives (0.1272), the Number of People Participating in Medical Insurance for Urban and Rural Residents (0.1196), the Number of Village Committee (0.1033), the Total Electricity Consumed by The Rural Society (0.0917), and the Per Capita Disposable Income of Rural Households (0.0741) rank as the most important, while in terms of developing rural e-commerce, the Revenue from Postal Services (0.1985); Network Retail Sales (0.1216); E-commerce Transaction Volume (0.1202); the Information Transmission, Computer Services, and Software Industry (0.1168); and the Demonstrable Integration of E-Commerce into Rural Counties (0.1052) ranked as the most important. In summary, this study found that the coupling coordination degree between rural revitalization and rural e-commerce in Hunan Province is jointly influenced by factors such as policy, economy, society, and human resources. In view of this, seven explanatory variables were selected to construct a spatial econometric model, including government support, infrastructure development, agricultural industry structure, digitalization level, human capital, regional consumption level, and logistics system. Each variable indicator is described as follows:

(1) Government support factor (GOV)—The government is able to facilitate the continuous growth of rural e-commerce utilizing policy orientation, financial subsidies, and tax incentives. The successful experience of the "Comprehensive Demonstration of E-commerce in Rural Areas" has proven that government support has facilitated the development of rural e-commerce and rural revitalization [13]. This factor is measured by the proportion of local fiscal expenditure to GDP.

(2) Infrastructure development factor (INF)—Infrastructure comprises the various types of basic facilities and services that support the normal functioning of social and economic activities in rural areas, and excellent infrastructure is the cornerstone of rural revitalization and rural e-commerce, providing more opportunities for farmers to participate in the modern economy. Huang et al. [47] showed empirically that highway construction can improve population flow and the movement of goods, which reduces the urban–rural revenue gap. Rural revitalization needs high-quality urban and rural highway construction for support, as the whole city highway route mileage is used to characterize it.

(3) Agricultural industry structure factor (AGR)—For one thing, rural e-commerce offers a novel development path for the agro-industry; for another, the upgrading of the agro-industrial structure boosts the prosperity of rural industries and creates a broader e-commerce market. For example, "Geographical indications" brand building for rural revitalization is not only a key initiative to promote the structural reform of the agricultural supply side but also an essential means to significantly boost the development of rural e-commerce [48]. The value of the total output of the agricultural, forestry, animal husbandry, and fishery industries in the city as a share of GDP is used as a measure.

(4) Digitalization level factor (DIG). Rural e-commerce relies on digital technology, while the development of digital agriculture and digital villages is an impetus for rural revitalization. Therefore, it is crucial to improve the level of rural digitization in order to promote both rural revitalization and rural e-commerce. The number of Internet broadband subscribers and the number of cell phone subscribers are selected to reflect this, and the entropy method is further used to measure the digitization level of each prefecture-level city [49].

(5) Human capital factor (HUM)—High-quality labor can provide professional skills and improve the use of technological equipment, management services, and market strategies; it is the primary resource driving rural revitalization and rural e-commerce development. This factor is measured using the number of college students enrolled per 10,000 people.

(6) Regional consumption level factor (CONSP)—The consumption expenditure of regional residents can reflect the consumption level and market demand of residents in the region [50], which boosts the evolution of rural e-commerce and rural revitalization from the demand side. This factor is expressed as the per capita consumption expenditure of rural residents.

(7) Logistics system factor (LOG)—Logistics is an essential link in the operation of rural e-commerce and is also an effective way to enhance the efficiency of agricultural product circulation and increase farmers' incomes [51]. This factor is measured by taking the number of units of the transportation, warehousing, and postal industries.

### 4.6.2. Analysis of Driver Regression Results

The spatial econometric regression results are shown in Table 9. Firstly, the LM test was conducted according to the judgment criteria of Anselin [52] to select an appropriate spatial econometric model. The results show that LM Error, LM Lag, and Robust-LM Lag all passed the significance test, while the Robust-LM Error failed the significance test; based on this, it was judged that the LM test with a lag term is significant and that the building of a spatial lag model (SAR) was required. Secondly, the Hausman test significantly rejected the original hypothesis, indicating the need to consider fixed effects. Fixed effects are classified as time-fixed, space-fixed, and time–space-fixed in spatial panel data, and after analyzing the types of fixed effects, the results show that the spatial lag model with time–space effects is the optimal model.

**Table 9.** Results of spatial econometric regression.

| Test | LM-Value | *p*-Value |
|---|---|---|
| LM Error | 55.091 | 0.000 |
| Robust-LM Error | 0.984 | 0.321 |
| LM Lag | 95.639 | 0.000 |
| Robust-LM Lag | 41.532 | 0.000 |
| Hausman test statistic | 31.77 | 0.000 |

By analyzing the regression results of the explanatory variables in the spatial lag model (Table 10), we found that infrastructure development, digitalization level, human capital, and regional consumption level all have a significant driving effect on the coupling coordination development of rural revitalization and rural e-commerce in Hunan Province, whereas the roles of governmental support, agricultural industry structure, and logistic level are not obvious.

**Table 10.** Results of driver regressions.

| Driver Factor | Coefficient | Standard Error | *p*-Value |
|---|---|---|---|
| Z-GOV | 0.0084615 | 0.92 | 0.355 |
| Z-INF | 0.3765986 *** | 3.07 | 0.002 |
| Z-AGR | −0.0004214 | −0.05 | 0.962 |
| Z-DIG | 0.0597762 *** | 5.33 | 0.000 |
| Z-HUM | 0.0810349 *** | 2.82 | 0.005 |
| Z-CONSP | −0.0064422 * | −1.79 | 0.073 |
| Z-LOG | 0.0049175 | 1.28 | 0.201 |

Note: The superscripts *** and * indicate 1% and 10% significance levels, respectively.

(1) Government support: It was generally agreed that governments would be able to provide policy support and environmental backing both for rural revitalization and rural e-commerce. It is worth noting that giving full weight to the determining function of the market in resource allocation and providing better governance are requirements for realizing high-quality socio-economic development. If there is no boundary system for government support, financial subsidies are distorted, and the management of investment financing is lax, it may lead to the abnormal development of government-supported projects. This may be an important reason for the insignificant results.

(2) Infrastructure development: Rural infrastructure is a guarantee and prerequisite for farmers' lives and agricultural production, as well as a cornerstone of rural development; it has a significant bearing on the achievement of the rural revitalization strategy. In particular, for the development of rural e-commerce, rural road access projects and the construction of information and communication infrastructure need to take place. Therefore, enhancing rural public infrastructure construction and upgrading public service facilities can provide a concrete guarantee for the harmonious development of rural revitalization and rural e-commerce.

(3) Agricultural industrial structure: It is difficult for the agricultural industry structure to influence the coordinated growth of rural revitalization and rural e-commerce. For this reason, the lack of support for digital technology constrains the development of digital technology-enabled agriculture, so it is difficult to achieve the sustainable upgrading of the agricultural structure. For one thing, the inadequacy of digital technology is mainly reflected in the singularity of technological applications and the lack of access to highly sophisticated digital technologies, resulting in a blockage of the process of digital transformation in agriculture. For another thing, although big data technology is at the core of the development of digital agriculture, data sharing may lead to problems such as ambiguous data property rights and privacy breaches, as well as the fact that when sharing agricultural data, farmers may encounter a series of problems, such as the opaque terms of data licenses, the ambiguity of data ownership, privacy issues, inequalities in bargaining power, and a lack of benefit sharing with farmers, which have hampered the development of agricultural industries driven by digital technology. As a result of these reasons, the coordinated development of rural revitalization and rural e-commerce is affected, which leads to non-significant estimation results.

(4) Digitalization level: Enhancing rural digitization levels can sustainably contribute to the growth of new forms of businesses in the rural digital economy. For one thing, rural e-commerce is used as a bond to access all kinds of resources serving the countryside; for another thing, it is used to allow primary, secondary, and tertiary industries to establish complementary advantages and integrated development with digital technology, which will become a new business mode for rural digital economies. Therefore, it is important to enhance the digitalization level in rural regions to effectively drive rural revitalization and rural e-commerce.

(5) Human capital: Human capital is at the core of rural e-commerce development and is the foundation and support for the realization of rural revitalization and high-quality rural development [53]. Human capital provides technical support and advanced management means for rural e-commerce, while talents with innovative entrepreneurial spirit are able to drive the development of emerging industries in the countryside and inject a lasting power source for rural revitalization.

(6) Regional consumption level: There is a negative correlation between regional consumption levels and the coordinated development of rural revitalization and rural e-commerce. When a region's consumption level is low, it may mean that the residents of the region have an insufficient consumption capacity, leading to rural e-commerce facing insufficient market demand, which affects the growth of rural e-commerce. Meanwhile, when residents' consumption level is low, it may mean that consumers in the region have a limited demand for high-value-added products or services, which constrains the emergence of rural industrial upgrading and new business models. Therefore, under such circumstances, the regional consumption level is negative.

(7) Logistics system: Inadequate logistics infrastructures may be an important factor contributing to the insignificant results of the estimation. For example, there may be a shortage of high-quality roads, warehouse facilities, and transportation means, which may lead to logistics inefficiencies, thus restricting the coordinated development of rural e-commerce and rural revitalization.

## 5. Discussion

This study explores the level of coupling coordination and driving factors of rural revitalization and rural e-commerce in Hunan Province in the context of the digital economy. In terms of the coupling coordination degree trend, it can be observed that the findings of this study are similar to those of Feng and Zhang [21]. Feng and Zhang examined 10 rural revitalization-demonstrating counties in Guizhou Province and discovered that the coupling coordination degree between rural revitalization and rural e-commerce was generally on the rise but that the coupling coordination degree was generally low. The reason for this is that Guizhou Province has always been one of the poorest regions in China, and its main focus has been on poverty alleviation, which has led to the neglect of rural e-commerce development in various areas of Guizhou. In contrast, this study observed an overall increasing trend in the coupling coordination between rural revitalization and rural e-commerce in the 14 prefectural-level cities in Hunan Province, but this study concluded that the coupling coordination degree in Hunan Province is getting better and better, gradually shifting from dysfunctional to coordinated. The main reason for this is the consistent emphasis on developing rural e-commerce in Hunan Province while promoting rural revitalization. The difference in the coupling coordination degree between the two provinces mainly stems from the fact that the economic foundation of each province is dissimilar. Compared to Guizhou Province, Hunan Province has a better rural economy, with better rural infrastructure and higher digitization levels, which gives it a greater advantage in the development of rural e-commerce. Therefore, this study believes that the level of socio-economic development of the region will have an impact on the coupling coordination degree between rural revitalization and rural e-commerce and that the coupling coordination degree between the two subsystems will increase with the development of the socio-economy.

In terms of the spatial distribution of the coupling coordination degree between rural revitalization and rural e-commerce, this study argues that regions with superior socio-economic conditions show high–high agglomeration, while, conversely, those with inferior socio-economic conditions show low–low agglomeration. And Liu et al. [2] similarly argued that socio-economic conditions significantly influence the spatial agglomeration of rural e-commerce. Therefore, as a matter of practice, it is important not only to develop regions with favorable socio-economic conditions in a high-quality manner but also to

effectively enhance the quality of economic development in economically underdeveloped regions so as to promote the synergistic development of the whole society.

There is a dearth of research in the literature exploring what drives the coupling coordination of rural revitalization and rural e-commerce. In turn, this study explored the factors that drive the coupling coordination degree between rural revitalization and rural e-commerce using observations from 14 prefectural-level cities in Hunan Province from 2013 to 2021. However, due to data unavailability, this study estimated missing data for certain years or regions, meaning that the results of this study possibly deviate from the actual results. Subsequent studies need to seek more comprehensive and valid data to make the results more realistic and reliable. Furthermore, this study concludes that infrastructure development, digitalization level, human capital, and regional consumption level significantly influence the coordinated development of rural revitalization and rural e-commerce, while the roles of government support, agricultural industry structure, and logistics level are not yet significant. This conclusion only applies to Hunan Province, and further studies should conduct research with a wider scope of application so that more regions can develop rural e-commerce to help rural revitalization.

## 6. Conclusions and Suggestions

### 6.1. Conclusions

Based on the panel data of 14 prefecture-level cities in Hunan Province from 2013 to 2021, this study focused on the spatio-temporal evolution characteristics of the coupling coordination development of rural revitalization and rural e-commerce and its driving factors using the entropy value method, a coupling coordination degree model, the Thiel index, the Moran index, and a spatial econometric model. Our major conclusions are as follows:

Firstly, as a whole, the level of coupling coordination between rural revitalization and rural e-commerce in Hunan Province has gradually increased and shifted gradually from dysfunction to coordination. From a regional perspective, the four major regions are the Chang-Zhu-Tan Area > Southern Hunan Area > Dongting Lake Area > Western Hunan Area. From a municipal perspective, the coupling coordination level of Changsha is in the absolute leading state, while the coupling coordination levels of Xiangxi and Zhangjiajie have been lagging behind for a long time. As a result, this study discovered that the coupling coordination degree is high in regions with good regional economic conditions, such as Changsha, while the coupling coordination degree is low in regions with poor regional economic conditions, such as Xiangxi and Zhangjiajie, which is due to the mismatch between the speed of rural e-commerce development and the speed of rural revitalization development. Thus, the support and guidance for the development of lagging regions need to be strengthened in order to reduce regional disparities.

Secondly, regional differences exist in the level of coupling coordination between rural revitalization and rural e-commerce in Hunan Province, but the overall differences are showing a narrowing trend. The differences among the four major economic regions mainly originate from intra-regional differences, with the order of regional differences and contribution rates being as follows: Chang-Zhu-Tan Area > Western Hunan Area > Dongting Lake Area > Southern Hunan Area. As a result, this study argues that differentiated measures should be implemented according to regional characteristics and that coordinated development within the region should be strengthened in order to improve the overall level of development.

Thirdly, there is an obvious spatial agglomeration feature in the coupling coordination of rural revitalization and rural e-commerce in Hunan Province, and it is mainly dominated by HH-type agglomeration and LL-type agglomeration, of which HH-type agglomeration areas are mainly concentrated in the eastern part of Hunan Province, and the LL-type agglomeration areas are concentrated in Zhangjiajie and Xiangxi in the western part of Hunan Province. As a result, one recommendation is to strengthen the development advantages of the eastern part of Hunan Province and address the development needs

of the western part of Hunan Province; another recommendation is to make full use of the advantageous resources of each region to strengthen inter-regional cooperation and exchanges so as to promote coordinated development.

Fourthly, some factors jointly drive the coordinated development of rural revitalization and rural e-commerce, such as government support, infrastructure development, agricultural industry structure, digitization level, human capital, regional consumption level, and logistics systems. Infrastructure development, digitalization level, and human capital have positive driving effects, and regional consumption level has negative effects, while government support, agricultural industry structure, and logistics levels have insignificant effects. Based on the above-mentioned results, this study argues that there should be more investment in and support for infrastructure construction, digitalization level, and human capital through policy guidance and structural adjustment to enhance the regional consumption level, optimize the structure of the agricultural industry, and improve logistics levels, which will promote the coordinated development of rural revitalization and rural e-commerce.

*6.2. Suggestions*

Firstly, it is important to emphasize regional development differences and strengthen intra-regional cooperation. For the Chang-Zhu-Tan Area, which has the biggest difference in regional development and is in the HH-type agglomeration area, we should emphasize the role of Changsha in radiating and driving the neighboring areas and give priority to the function of industrial clustering in the provincial capital city. For the Western Hunan Area, which has large differences in regional development and is in an LL-type agglomeration, we should utilize the latecomer advantages and catch-up effects of the cities of Zhangjiajie and Xiangxi in order to narrow the low degree of coordination due to the low level of rural revitalization and the low level of rural e-commerce. For the southern Hunan Area and Dongting Lake Area, which have small regional differences, on the one hand, according to regional characteristics, we should not only optimize the industrial structure within the region and develop differentiated industries but also strengthen the iterative development of industries to promote economic transitions and upgrading in rural areas. On the other hand, we should enhance intra-regional cooperation and inter-provincial and municipal cooperation. The industry in the Southern Hunan Area could dock with the industries in Guangdong, Hong Kong, and Macao, and the industry in Dongting Lake Area could dock with the industries in the Yangtze River Basin Economic Belt, which would further promote the simultaneous enhancement of rural revitalization level and rural e-commerce level.

Secondly, expanding current investment in agricultural and rural infrastructure and promoting the upgrading of rural infrastructure are also needed. A well-connected infrastructure is a necessary condition for realizing rural revitalization and promoting the development of rural e-commerce in the new era. One is demand-oriented, focusing on investment in agricultural and rural infrastructure shortcomings, highlighting the construction of modern agricultural facilities, rural roads, rural Internet penetration, agricultural preservation and cold chain logistics facilities, and other projects. We should strive to solve the imbalances and insufficiencies in rural infrastructure construction so that the smooth operation of rural e-commerce can be guaranteed. Another suggestion is to promote the role of the government in planning guidance, policy support, and organizational safeguards, which would not only facilitate the modernizing of rural infrastructure but also fully safeguard the rights of farmers. At the same time, it is necessary to avoid supply shortages and mismatches in order to effectively increase the quality and level of rural infrastructure supply.

Thirdly, we need to enhance digitalization and facilitate the development of the rural digital economy. For one thing, we should keep broadening the scope of digital technology application in rural regions and strengthening the embedding and application of digital means in the whole agricultural industry chain. Utilizing digital technology to empower the entire process of agricultural planting, breeding, and animal husbandry promotes the

digitization of rural industries. Another suggestion is to explore new methods of assisting agriculture through digitalization. For example, we could promote the transboundary integration of e-commerce, modern logistics, and agriculture; push for the standardized development of modes such as "e-commerce to help farmers" and "live broadcasting with agricultural goods"; and foster new forms of agricultural e-commerce. Broadening the agricultural product sales channels and upgrading rural production and operation digitalization through the above-mentioned ways will contribute significantly to increasing farmers' incomes.

Fourthly, we need to improve skills training for farmers and accelerate the growth of rural e-commerce talents. This would involve actively conducting e-commerce training activities in rural areas to train a group of high-quality farmers who understand e-commerce. Through e-commerce training, farmers can make use of the e-commerce platform to achieve the purpose of "promoting commodities and obtaining sales channels", which would help to promote rural revitalization and increase farmers' incomes. There is also a need to improve the entrepreneurial support policies on rural e-commerce and actively encourage all kinds of people to use e-commerce to start their own business, as this could give rise to several local e-commerce talents. We need to encourage universities and colleges around the country to establish a talent training cooperation mechanism with e-commerce enterprises and set up an enterprise training base to cultivate comprehensive talents who are familiar with all aspects of e-commerce entrepreneurship.

**Author Contributions:** Conceptualization, C.Z. writing—original draft preparation, C.Z.; writing—review and editing, C.Z. and W.L. All authors have read and agreed to the published version of the manuscript.

**Funding:** Project "An Empirical Study on the Coupling Coordination and Driving Factors of Rural Revitalization and Rural E-Commerce in the Context of the Digital Economy: The Case of Hunan Province, China" of Postgraduate Scientific Research and Innovation Project of the Department of Postgraduate Education and Teaching, Hunan University of Humanities, Science and Technology, Project No.: ZSCX2023Y11.

**Institutional Review Board Statement:** Not applicable.

**Informed Consent Statement:** Not applicable.

**Data Availability Statement:** The data presented in this study are available upon request from the corresponding author.

**Conflicts of Interest:** The authors declare no conflicts of interest.

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
