# Peer review of "An Empirical Study on the Coupling Coordination and Driving Factors of Rural Revitalization and Rural E-Commerce in the Context of the Digital Economy: The Case of Hunan Province, China"

_sustainability, doi:10.3390/su16093510_

Round 1

Reviewer 1 Report

Comments and Suggestions for Authors

Comments:

Manuscript ID: sustainability-2932602

Title: An Empirical Study on the Coupling Coordination and Driving Factors of Rural Revitalization and Rural E-commerce in the Context of Digital Economy: The Case of Hunan Province of China

Section: Economic and Business Aspects of Sustainability

Special Issue: Experience Design and Digital Transformation in Business 

The topic is interesting, the authors have made an interesting assessment, which is attractive to relevant researcher to some extent. The subject of reveals the relationship between the coordinated development of rural revitalization and rural e-commerce fits in the general scope of sustainability. Just, I suggest some modifications to further improve the quality of the article.

Abstract:

The background and method are too redundant, and the conclusion part is too little, and the elaboration is not clear enough.

Introduction and Literature Review:

The last sentence of the second paragraph of the introduction is too long and the meaning is not clear and concise.

The literature review has not sufficiently summarized the existing relevant research, thus weakening the necessity of the research question.

3. Construction of the evaluation index system and 4. Research Methods:

This part seems to lack the explanation and map of the study area.

It is proposed that parts 3 and 4 be merged into one part,named 3Materials and Methods, title 3.1 is thestudy area and data source, title 3.2 is Methodology, Construction of the evaluation index system is also in Part 3. This part is not clear and should be organized to introduce the methods used in this study(Includes the methods used in the current Part 6).

Empirical Analysis:

There should have figures in the analysis results in this section(such as Region Difference in the Coupling Coordination Degree of Rural Revitalization and Rural E-commerce can use a figure to express it more vividly), however, from 5.1 to 5.5 there is not a single figure.

Currently Part 6 (“6. Analysis of Driving Factors for Coupled and Coordinated Development of Rural Revitalization and Rural E-commerce in Hunan Province”) is proposed to be merged into this section. Because it's part of empirical research.

Discussion: This article lacks discussion section. This paper maybe fail to engage with the wider readership of Sustainability. Any Description and discussions should be beyond the local case itself, otherwise it cannot attract more international readers. It is suggested to add sub-headings to clearly dissect the topic of discussion and make substantial revisions before re-examination.

Conclusions and Suggestions: In general, the conclusion should be based on the previous results and discussions, and should be brief and concise, giving only the core concluding content without elaboration.It is suggested that the content of the conclusion should be refined, so far it is more like the result.

It would be better if the authors can supplement the research contributions, and recommendations of future studies.

Overall, the paper is a good technical effort. There still are some repetition and grammar errors in the MS, authors need to check them carefully.

Comments on the Quality of English Language

Overall, the paper is a good technical effort. There still are some repetition and grammar errors in the MS, authors need to check them carefully. 

Author Response

Point 1: In abstract,the background and method are too redundant, and the conclusion part is too little, and the elaboration is not clear enough.

Response 1:

We have streamlined the background and methods section in the abstract, as well as supplemented the conclusions section.

The coupling coordination development of rural revitalization and rural e-commerce is of great significance in promoting the economic growth of rural areas. This study takes the observations of 14 prefecture-level cities in Hunan Province from 2013 to 2021 as the research samples, and adopts the methods of coupling coordination model and spatial econometric model to analyze the spatio-temporal evolution characteristics and driving factors of the coupling coordination development of rural revitalization and rural e-commerce in Hunan Province. The findings indicate that: (1) The degree of coupling coordination between rural revitalization and rural e-commerce in Hunan Province has gradually been increased, showing a transition path from dissonance to coordination. (2) There are regional differences in the degree of coupling coordination between rural revitalization and rural e-commerce in Hunan Province, with the overall difference showing a fluctuating trend of narrowing, and the intra-regional difference is the main reason for the overall difference. (3) The degree of coupling coordination shows spatial clustering characteristics, with high-high clustering areas mainly concentrated in the eastern part of Hunan Province and low-low clustering areas concentrated in the western part of Hunan Province. (4) There is a notable driving effect of infrastructure establishment, digitalization level, human capital, and regional consumption level on the coupling coordination of rural revitalization and rural e-commerce in Hunan Province. Above all, it provides effective suggestions for promoting the coupling coordination of rural revitalization and rural e-commerce.

Point 2: The last sentence of the second paragraph of the introduction is too long and the meaning is not clear and concise.

Response 2

Thank you for your suggestion. We have adjusted the last sentence of the second paragraph of the introduction to make the study purpose and study value more clear and concise.

It is important to clarify the coupling coordination relationship between rural revitalization and rural e-commerce in Hunan Province, as well as to identify the driving factors that promote the coordinated development of rural revitalization and rural e-commerce, which will be of great reference value for promoting rural economic development and realizing rural revitalization in Hunan Province.

Point 3: The literature review has not sufficiently summarized the existing relevant research, thus weakening the necessity of the research question.

Response 3:

Thank you for your suggestion. We have fully revised the literature review to reinforce the necessity of the research question.

First, this study explains and reviews the objectives, requirements, and functions of rural revitalization. Second, this study explains and reviews the definition and functions of rural e-commerce. Last, this study reviews the progress of research related to the integrated exploration of rural revitalization and rural e-commerce.

Through the above analysis, this study found out the shortcomings of the existing literature: (1) Research yardsticks are mostly focused on the macro level, such as national and provincial levels, with fewer research results at the municipal level. However, compared with the macro level, the municipal area, as an important geographical unit at the small level, is more likely to develop rural e-commerce and realize rural revitalization; (2) There is a lack of empirical research on the spatio-temporal evolution and regional differences in the coupling coordination relationship between rural revitalization and rural e-commerce in Hunan Province, and it is rare to explore the characteristics of the spatio-temporal evolution and driving factors of the coupling coordination relationship between rural revitalization and rural e-commerce in the context of the digital economy. In view of this, this study makes the necessity of the research question even clearer.

Point 4: This part seems to lack the explanation and map of the study area.

Response 4:

Thank you for your suggestions. We have added explanation and map of the study area.

This study is based on 14 prefectural-level cities in Hunan Province which are divided into four major economic sectors according to the Hunan Provincial National Economic and Social Development Statistical Bulletin, including Chang-Zhu-Tan Area, Southern Hunan Area, Western Hunan Area, and Dongting Lake Area. Among them, Chang-Zhu-Tan Area refers to the 3 cities of Changsha, Zhuzhou and Xiangtan, Southern Hunan Area refers to the 3 cities of Hengyang, Chenzhou and Yongzhou, Western Hunan Area refers to the 5 cities ( autonomous prefecture) of Shaoyang, Zhangjiajie, Huaihua, Loudi and Xiangxi , and Dongting Lake Area refers to the 3 cities of Yueyang, Yiyang and Changde. The map is shown below.

 Map of Hunan Province

Point 5: â‘ It is proposed that parts 3 and 4 be merged into one part,named 3"Materials and Methods", title3.1 is the "study area and data source", title 3.2 is "Methodology", “Construction of the evaluation index system” is also in Part 3. This part is not clear and should be organized to introduce the methods used in this study(Includes the methods used in the current Part 6).

â‘¡Currently Part 6 ("6. Analysis of Driving Factors for Coupled and Coordinated Development of Rural Revitalization and Rural E-commerce in Hunan Province”) is proposed to be merged into Empirical Analysis section. Because it's part of empirical research.

Response 5

We have adjusted the structure of the manuscript based on your valuable comments. The adjusted parts are as follows:

Part 3 Materials and Methods

3.1 Study Area and Data Source

3.1.1 Study Area

3.1.2 Data Source

3.2 Methodology

3.2.1 Entropy method

3.2.2 Coupling Coordination Degree Model

3.2.3 Thiel Index

3.2.4 Moran Index

3.2.5 Spatial Econometric Model

3.3 Construction of the evaluation index system

3.3.1 Construction of Evaluation Index System for Rural Revitalization in Hunan Province

3.3.2 Construction of Evaluation Index System for Rural E-commerce in Hunan Province

Part 4 Empirical Analysis

4.1 Rural Revitalization Development Index

4.2 Rural E-commerce Development Index

4.3 Analysis of the coupling coordination between rural revitalization and rural e-commerce

4.4 Region Difference in the Coupling Coordination Degree of Rural Revitalization and Rural E-commerce

4.5 Spatial Distribution of Coupling Coordination Degree of Rural Revitalization and Rural E-commerce

4.6 Analysis of Driving Factors for Coupled and Coordinated Development of Rural Revitalization and Rural E-commerce in Hunan Province

Point 6: In empirical analysis, there should have figures in the analysis results in this section(such as “Region Difference in the Coupling Coordination Degree of Rural Revitalization and Rural E-commerce" can use a figure to express it more vividly), however, from 5.1 to 5.5 there is not a single figure.

Response 6:

Thank you for your suggestion. We tried our best to add figure in section 4.3 (Analysis of the coupling coordination between rural revitalization and rural e-commerce). The figure chapter shows the degree of coupling coordination for the years 2013, 2017, and 2021 so that the changes that occurred in the degree of coupling coordination during the study period can be better observed. As shown in the figure below:

 Degree of coupling coordination between rural revitalization and rural e-commerce in Hunan Province. (a) 2013 (b) 2017 (c) 2021

Point 7: This article lacks discussion section. This paper maybe fail to engage with the wider readership of Sustainability, Any Description and discussions should be beyond the local case itself, otherwise it cannot attract more international readers. it is suggested to add sub-headings to clearly dissect the topic of discussion and make substantial revisions before re-examination.

Response 7:

Thank you for your suggestions. We have added a discussion section.

This study compares with the research results of Feng & Zhang and Liu el at. and it is found that the level of regional socio-economic development affects the degree of coupling coordination and spatial dispersion of rural revitalization and rural e-commerce.In addition, this study adds to the drivers of the coupling coordination of rural revitalization and rural e-commerce. However, the relevant drivers apply only to Hunan Province, and further studies should explore a conclusion with a wider scope of application so that more regions can develop rural e-commerce to help rural revitalization.

Point 8: Conclusions and Suggestions: In general, the conclusion should be based on the previous results and discussions, and should be brief and concise, giving only the core concluding content without elaboration.It is suggested that the content of the conclusion should be refined, so far it is more like the result.

Response 8

Thank you for your suggestions. We have revised the conclusions section to make them more granular.

This study provides a rule in accordance with the coupling coordination change of rural revitalization and rural e-commerce in Hunan Province based on the results of the study.

Point 9: It would be better if the authors can supplement the research contributions, and recommendations of future studies.

Response 9

Thank you for your suggestions. We've incorporated suggestions for future research into the discussion section.

This study highlights that future exploration of the driving factors should provide a conclusion with a broader scope of application.

Reviewer 2 Report

Comments and Suggestions for Authors

Overall Review:The manuscript selects the panel data of 14 prefecture-level cities in Hunan Province from 2013 to 2021, and applies the entropy method, the coupled coordination degree model, the Thiel index, the Moran index and the spatial measurement model to analyse the "rural revitalization" and "rural e-commerce" respectively "development level", and studied the spatio-temporal evolution characteristics of the coupled and coordinated development of rural revitalisation and rural e-commerce and their driving factors. The manuscript study is universal, with clear regulations and tight logic in the research and analysis. However, the manuscript seems to have some problems worth thinking about. On the one hand, the content of the manuscript still has errors and lack of clarity in some upper formats (refer to the specific issues); on the other hand, the manuscript is doubtful in some analyses of the research content analysis. The specific issues are listed below:

P1.Line19-20  "ABSTRACT" results section does not seem to summarise the findings of the manuscript very well and could be considered for revision.

P1. Line23  "Introduction" This section does not seem to have a statement of the significance of the study and the problem, consider checking for changes.

P2. Line65  The author of "Rural Revitalization" may want to explain and review rural revitalisation, but the body of the text is mostly in the expression of "digital countryside", which seems to have a lack of linkage with the research content of the manuscript, so in order to make the content of the article clearer, it can be considered to review the progress of research related to "rural revitalization", which will lead to "rural e-commerce".

P2-3. Line90-93  Affirmative conceptualisation, seems to lack reference, may consider checking. (Similar problems are found throughout the rest of the manuscript, which may be considered for examination in its entirety.)

P3. Line113-115  Affirmative statements, which seem to lack reference, may be considered for inspection.

P4. Line148-158  Seems to lack a summary of the research questions and could be considered for revision.

P5. Line204-220  These statements may have been redundant when the authors were working on the construction of indicators for the evaluation of "rural e-commerce", and a general overview could be provided to consider checking for changes.

P8. Line269   There is a lack of description of the values of α and β in the coupled coordination degree model, which should generally be described, for example, how the values were taken.

P19. Line621-683  In conducting the driver analysis, it seems that instead of analysing the analysis against the previous study, the factors are outlined and there is no mention of the drivers in the Hunan Province, where the study case is located, so consider checking the revision and conducting the driver analysis against the study case.

Author Response

Point 1: Line19-20 "ABSTRACT" results section does not seem to summarise the findings of the manuscript very well and could be considered for revision.

Response 1

We have streamlined the background and methods section in the abstract, as well as supplemented the conclusions section.

The coupling coordination development of rural revitalization and rural e-commerce is of great significance in promoting the economic growth of rural areas. This study takes the observations of 14 prefecture-level cities in Hunan Province from 2013 to 2021 as the research samples, and adopts the methods of coupling coordination model and spatial econometric model to analyze the spatio-temporal evolution characteristics and driving factors of the coupling coordination development of rural revitalization and rural e-commerce in Hunan Province. The findings indicate that: (1) The degree of coupling coordination between rural revitalization and rural e-commerce in Hunan Province has gradually been increased, showing a transition path from dissonance to coordination. (2) There are regional differences in the degree of coupling coordination between rural revitalization and rural e-commerce in Hunan Province, with the overall difference showing a fluctuating trend of narrowing, and the intra-regional difference is the main reason for the overall difference. (3) The degree of coupling coordination shows spatial clustering characteristics, with high-high clustering areas mainly concentrated in the eastern part of Hunan Province and low-low clustering areas concentrated in the western part of Hunan Province. (4) There is a notable driving effect of infrastructure establishment, digitalization level, human capital, and regional consumption level on the coupling coordination of rural revitalization and rural e-commerce in Hunan Province. Above all, it provides effective suggestions for promoting the coupling coordination of rural revitalization and rural e-commerce.

Point 2: Line23 "Introduction" This section does not seem to have a statement of the significance of the study and the problem,consider checking for changes.

Response 2:

Thank you for your suggestions. We have revised the introduction to make the significance of the study and questions of the study clearer.

Study problem: In terms of the national level, the development speed of rural e-commerce in Hunan Province has a certain gap compared with that of the eastern coastal cities. Therefore, it is essential to strengthen the research on how to maintain the existing rural e-commerce market and further expand the rural e-commerce market in Hunan Province.

Study Significance: It is important to clarify the coupling coordination relationship between rural revitalization and rural e-commerce in Hunan Province, as well as to identify the driving factors that promote the coordinated development of rural revitalization and rural e-commerce, which will be of great reference value for promoting rural economic development and realizing rural revitalization in Hunan Province.

Point 3: â‘ Line65 The author of "Rural Revitalization" may want to explain and review rural revitalisation, but the body of the text is mostly in the expression of "digital countryside" which seems to have a lack of linkage with the research content of the manuscript. so in order to make the content of the article clearer, it can be considered to review the progress of research related to "rural revitalization", which will lead to "rural e-commerce".

â‘¡Line148-158 Seems to lack a summary of the research questions and could be consideredfor revision.

Response 3:

Thank you for your suggestion. We have fully revised the literature review to reinforce the necessity of the research question.

First, this study explains and reviews the objectives, requirements, and functions of rural revitalization. Second, this study explains and reviews the definition and functions of rural e-commerce. Last, this study reviews the progress of research related to the integrated exploration of rural revitalization and rural e-commerce.

Through the above analysis, this study found out the shortcomings of the existing literature: (1) Research yardsticks are mostly focused on the macro level, such as national and provincial levels, with fewer research results at the municipal level. However, compared with the macro level, the municipal area, as an important geographical unit at the small level, is more likely to develop rural e-commerce and realize rural revitalization; (2) There is a lack of empirical research on the spatio-temporal evolution and regional differences in the coupling coordination relationship between rural revitalization and rural e-commerce in Hunan Province, and it is rare to explore the characteristics of the spatio-temporal evolution and driving factors of the coupling coordination relationship between rural revitalization and rural e-commerce in the context of the digital economy. In view of this, this study makes the necessity of the research question even clearer.

Point 4: Line90-93 Affirmative conceptualisation, seems to lack reference, may consider checking (Similar problems are found throughout the rest of the manuscript, which may be considered for examination in its entirety.)

Response 4

Thank you for your kind reminder. We have supplemented the literature source. It comes from the definition of rural e-commerce on the Chinese government website.

Point 5: Line113-115 Affirmative statements, which seem to lack reference, may be considered for inspection.

Response 5

Thank you for your kind reminder. We have supplemented the literature source.It comes from the Digital Agriculture Report jointly released by Zhejiang University (ZJU) and the Food and Agriculture Organization of the United Nations (FAO).

Point 6: Line204-220 These statements may have been redundant when the authors were working on the construction of indicators for the evaluation of "rural e-commerce" and a general overview could be provided to consider checking for changes.

Response 6

Thank you for your suggestions. We have completely revised the construction section of the rural e-commerce evaluation index.

According to the external environment mentioned for the development of rural e-commerce in Hunan Province, it selected Information and Communication Technology, Logistics Services, Degree of Economic and Social Development, and Business Environment as the first-level indexes, from which it extracted 16 second-level indexes reflecting the current situation of the development of rural e-commerce in Hunan Province, so as to construct a rural e-commerce evaluation indicator system.

Point 7: Line269 There is a lack of description of the values of α and β in the coupled coordination degree model, which should generally be described, for example, how the values were taken.

Response 7

Thank you for your kind reminder. We add the description of the values of α and β in the coupling coordination model.

This study assumes that rural revitalization and rural e-commerce are of the same importance. Therefore, the values of α and β are equally weighted and take the value of 0.5.

Point 8: Line621-683 In conducting the driver analysis. it seems that instead of analysing the analysis against the previous study, the factors are outlined and there is no mention of the drivers in the Hunan Province, where the study case is located, so consider checking the revision and conducting the driver analysis against the study case.

Response 8

Thank you for your suggestion. We have refined the driving factor selection section.

In order to seize the driving factors of coupling coordination development of rural revitalization and rural e-commerce in Hunan Province. On the one hand, it combs through the literature on the influencing factors of rural revitalization and rural e-commerce in recent years. It is found that resource endowment, economic location, policy factors, and infrastructure are the fundamental drivers affecting rural revitalization; While rural elites, Internet penetration, logistics, agricultural industry structure, and government support play an influential role in contributing to the construction of the rural e-commerce industry chain. On the other hand, based on the weights of each evaluation index measured in the previous section, it is found that in terms of realizing rural revitalization, the Number of Specialized Farmers' Cooperatives (0.1272), the Number of People Participating in Medical Insurance for Urban and Rural Residents (0.1196), the Number of Village Committee (0.1033), the Total Electricity Consumption by The Rural Society (0.0917), and the Per Capita Disposable Income of Rural Households (0.0741) rank in the top of importance; while in terms of developing rural e-commerce, the Revenue from Postal (0.1985), Network Retail Sales (0.1216), E-commerce Transaction Volume (0.1202), Information Transmission, Computer Services and Software Industry (0.1168), and E-Commerce into Rural Integrated Demonstration County (0.1052) ranked in the top of importance. In summary, it is argued that the degree of coupling coordination between rural revitalization and rural e-commerce in Hunan Province is compositely influenced by factors such as policy, economy, society and human resources. In view of this, there are 7 explanatory variables selected to construct a spatial econometric model, including government support, infrastructure, agricultural industry structure, digitalization level, human capital, regional consumption level, and logistics system.

Reviewer 3 Report

Comments and Suggestions for Authors

The paper presents a study on the relation between rural revitalization and e-commerce. The study was correctly planned and conducted. It shows the development of e-commerce in Hunan province and the differences among its different parts when it comes to e-commerce development.

Author Response

Thank you for your valuable time to comment on this study, I appreciate your support. In order to rationalize and improve the study, we have adjusted and revised some sections of the manuscript by combining the suggestions of the other two reviewers.

  1. In the abstract,we have streamlined the background and methods section in the abstract, as well as supplemented the conclusions section.
  2. In the introduction, We have revised the introduction to make the significance of the study and questions of the study clearer.
  3. In the literature review, We have fully revised the literature review to reinforce the necessity of the research question.
  4. In the materials and methods, first, we restructured the manuscript. We merged Part 3 (Construction of the evaluation index system) and Part 4 (Research Methods) and named them “3. Materials and Methods”. Second, we added a description of the study area. Last, we revised the corresponding descriptions of the construction of evaluation indicators for rural revitalization and rural e-commerce.
  5. In the empirical analysis, Part 6 ("6. Analysis of Driving Factors for Coupled and Coordinated Development of Rural Revitalization and Rural E-commerce in Hunan Province”) is merged into the empirical analysis section. Because it is a part of the empirical research.
  6. In the discussion, we have added a discussion section.
  7. In the conclusions and suggestions, we have revised the conclusions section to make them more granular.

Thank you once again. We hope that the adjustments will be equally supported by yours.

Round 2

Reviewer 1 Report

Comments and Suggestions for Authors

Comments:

Manuscript ID: sustainability-2932602

Title: An Empirical Study on the Coupling Coordination and Driving Factors of Rural Revitalization and Rural E-commerce in the Context of Digital Economy: The Case of Hunan Province of China

Section: Economic and Business Aspects of Sustainability

Special Issue: Experience Design and Digital Transformation in Business 

This revision manuscript (MS) is a great improvement on the previous. The work presented in the MS has scientific value. In particular, it analyze the spatio-temporal evolution characteristics and driving factors of the coupling coordination development of rural revitalization and rural e-commerce in Hunan Province, which will provide basic reference for rural planning. The subject fits in the general scope of Sustainability. The revision MS is more straightforward than the previous one, well written and easy to understand. Therefore, I'm glad to recommend accept it for publication after minor language and reference check.

Author Response

Dear reviewer:

Thank you very much for your professional review of the manuscript. We are very excited by your comments. As you were concerned, the language and references of the manuscript needed to be checked. According to your suggestions, we have checked the manuscript with the following changes.

For the language portion of the manuscript, we have made many efforts. Firstly, we did our best to check for grammar and spelling errors word by word. We have not listed the changes here, but have marked them in red in the revised manuscript. Secondly, we standardized the professional terminology used in the manuscripts. For example, change “Chang-Zhu-Tan Region” to “Chang-Zhu-Tan Area”; change “Southern Hunan Region” to “Southern Hunan Area”; change “Western Hunan Region” to “Western Hunan Area”; change “Dongting Lake Region” to “Dongting Lake Area”; change “inter-region” to “inter-regional”; change “intra-region” to “intra-regional”. Thirdly, in order to make the content of the manuscript clear, this study replaces “it” with “this study” where relevant.

For the reference checking of the manuscript, we compared the cited references one by one and made the corresponding changes. Firstly, we have deleted the 10th reference in the original manuscript. Because it is an unnecessary citation. Secondly, we added the 37th literature. We supplemented the literature on the selection of the Degree of Economic and Social Development as a first-level evaluation index for rural e-commerce. Thirdly, in order to make the cited content correspond to the corresponding references, we try to split the reference labeling of the merged citation to the corresponding position.

Last but not least, we have tried our best to improve the manuscript and made some red marked changes in the revised paper, but it will not affect the content and framework of the paper. We appreciate for Reviewers’ warm work earnestly, and hope the correction will meet with approval. Once again, thank you very much for your comments and suggestions.

Sincerely,

Canjiang Zhu

Reviewer 2 Report

Comments and Suggestions for Authors

Based on the comments of the reviewers, the author made a comprehensive revision to the abstract, introduction, research method, discussion and conclusion of the manuscript.

Author Response

(The authors gave the same response as above.)
